# Distributional Reinforcement Learning via Sinkhorn Iterations

## Abstract

Distributional reinforcement learning (RL) is a class of state-of-the-art algorithms that estimate the whole distribution of the total return rather than only its expectation. The representation manner of each return distribution and the choice of distribution divergence are pivotal for the empirical success of distributional RL. In this paper, we propose a new class of *Sinkhorn distributional RL (Sinkhorn-DRL)* algorithm that learns a finite set of statistics, i.e., deterministic samples, from each return distribution and then leverages Sinkhorn iterations to evaluate the Sinkhorn distance between the current and target Bellman distributions. Remarkably, Sinkhorn divergence interpolates between the Wasserstein distance and Maximum Mean Discrepancy (MMD). This allows our proposed SinkhornDRL algorithm to find a sweet spot leveraging the geometry of optimal transport based distance and the unbiased gradient estimates of MMD. Finally, experiments on the suit of 55 Atari games reveal the competitive performance of SinkhornDRL algorithm as opposed to existing state-of-the-art algorithms.

## 1 Introduction

Classical reinforcement learning (RL) algorithms are normally based on the expectation of discounted cumulative rewards that an agent observes while interacting with the environment. Recently, a new class of RL algorithms called *distributional RL* estimates the full distribution of total returns and has exhibited the state-of-the-art performance in a wide range of environments [2, 8, 7, 24, 26, 17].

From the literature of distributional RL, it is easily recognized that algorithms based on either Wasserstein distance or MMD have gained great attention due to their superior performance. As such, their mutual connection from the perspective of mathematical properties intrigues us to explore further in order to design new algorithms. Particularly, Wasserstein distance, long known to be a powerful tool to compare probability distributions with non-overlapping supports, has recently emerged as an appealing contender in various machine learning applications. It is known that Wasserstein distance was long disregarded because of its computational burden in its original form to solve an expensive network flow problem. However, recent works [21, 14] have shown that this cost can be largely mitigated by settling for cheaper approximations through strongly convex regularizers. The benefit of this regularization has opened the path to wider applications of the Wasserstein distance in relevant learning problems, including the design of distributional RL algorithms.

The Sinkhorn divergence [21] introduces the entropic regularization on the Wasserstein distance, allowing it tractable for the evaluation especially in high-dimensions. It has been successfully applied in numerous crucial machine learning developments, including the Sinkhorn-GAN [14] and Sinkhorn-based adversarial training [23]. More importantly, it has been shown that Sinkhorn divergence interpolates Wasserstein ditance and MMD, and their equivalence form can be well established in the limit cases [11, 18, 17]. However, a Sinkhorn-based distributional RL algorithm has not yet been

Submitted to 36th Conference on Neural Information Processing Systems (NeurIPS 2022). Do not distribute.

formally proposed and its connection with algorithms based on Wasserstein distance and MMD is also less studied. Therefore, a natural question is *can we design a new class of distributional RL algorithms via Sinkhorn divergence, thus bridging the gap between existing two main branches of distributional RL algorithms?* Moreover, the dominant quantile-based algorithms, e.g., QR-DQN [8], aimed at approximating Wasserstein distance, suffers from the non-crossing issue in the quantile estimation [26], while sample-based Sinkhorn algorithm can naturally circumvent this problem.

In this paper, we propose a novel distributional RL algorithm based on *Sinkhorn divergence*. Firstly, we point out the key roles of distribution divergence and representation of value distribution in the design of distributional RL. After a detailed introduction of our proposed SinkhornDRL algorithm, we theoretically analyze its convergence guarantee and moment matching behavior of distributional Bellman operators under Sinkhorn divergence. Thus, a regularized MMD equivalence form of Sinkhorn divergence is derived, interpreting the empirical success of our algorithms in real applications. Finally, we compare the performance of our SinkhornRL algorithm with typical baselines on 55 Atari games, verifying the competitive performance of our proposal. Our approach inspires researchers to find a trade-off that simultaneously leverages the geometry of the Wasserstein distance and the favorable unbiased gradient estimate property of MMD while designing new distributional RL algorithms in the future.

## 2 Preliminary Knowledge

### 2.1 Distributional Reinforcement Learning

In the classical RL setting, an agent interacts with an environment via a Markov decision process (MDP), a 5-tuple $(\mathcal{S}, \mathcal{A}, R, P, \gamma)$, where $\mathcal{S}$ and $\mathcal{A}$ are the state and action spaces, respectively. $P$ is the environment transition dynamics, $R$ is the reward function and $\gamma \in (0, 1)$ is the discount factor.

**From Value function to Value distribution.** Given a policy $\pi$, the discounted sum of future rewards is a random variable $Z^\pi(s, a) = \sum_{t=0}^{\infty} \gamma^t R(s_t, a_t)$, where $s_0 = s$, $a_0 = a$, $s_{t+1} \sim P(\cdot|s_t, a_t)$, and $a_t \sim \pi(\cdot|s_t)$. In the control setting, expectation-based RL is based on the action-value function $Q^\pi(s, a)$, which is the expectation of $Z^\pi(s, a)$, i.e., $Q^\pi(s, a) = \mathbb{E}[Z^\pi(s, a)]$. By contrast, distributional RL focuses on the action-value distribution, the full distribution of $Z^\pi(s, a)$, and the incorporation of additional distributional knowledge intuitively interprets its empirical success.

**Distributional Bellman operators.** For the policy evaluation in expectation-based RL, the action-value function is updated via the Bellman operator $\mathcal{T}^\pi Q(s, a) = \mathbb{E}[R(s, a)] + \gamma \mathbb{E}_{s' \sim p, \pi}[Q(s', a')]$. In distributional RL, the action-value distribution of $Z^\pi(s, a)$ is updated via the distributional Bellman operator $\mathfrak{T}^\pi$

$$\mathfrak{T}^\pi Z(s, a) = R(s, a) + \gamma Z(s', a'), \tag{1}$$

where $s' \sim P(\cdot|s, a)$ and $a' \sim \pi(\cdot|s')$. The equality in Eq. 1 implies that random variables of both sides are equal in distribution. The distributional Bellman operator $\mathfrak{T}^\pi$ is contractive under certain distribution divergence metrics, but the distributional Bellman optimality operator $\mathfrak{T}$ can only converge to a set of optimal non-stationary value distributions in a weak sense [9].

### 2.2 Divergences between Measures

**Optimal Transport (OT) and Wasserstein Distance** The optimal transport (OT) metric between two probability measures $(\mu, \nu)$ supported on two metric spaces is defined as the solution of the linear program:

$$\min_{\Pi \in \mathbf{\Pi}(\mu, \nu)} \int c(x, y) \mathrm{d}\Pi(x, y), \tag{2}$$

where $c$ is the cost function and $\Pi$ is the joint distribution with marginals $(\mu, \nu)$. Wasserstein distance (a.k.a. earth mover distance) is a special case of optimal transport with the Euclidean norm as the cost function. In particular, given two scalar random variables $X$ and $Y$, $p$-Wasserstein metric $W_p$ between the distributions of $X$ and $Y$ can be simplified as

$$W_p(X, Y) = \left( \int_0^1 \left| F_X^{-1}(\omega) - F_Y^{-1}(\omega) \right|^p d\omega \right)^{1/p}, \tag{3}$$

where $F^{-1}$ is the inverse cumulative distribution function of a random variable. The desirable geometric property of Wasserstein distance allows it to recover full support of measures, but it suffers from the curse of dimension [13, 1].

**Maximum Mean Discrepancy**    The squared Maximum Mean Discrepancy (MMD) $\text{MMD}_k^2$ with the kernel $k$ is formulated as

$$\text{MMD}_k^2 = \mathbb{E}\left[k\left(X, X'\right)\right] + \mathbb{E}\left[k\left(Y, Y'\right)\right] - 2\mathbb{E}\left[k(X, Y)\right], \tag{4}$$

where $k(\cdot, \cdot)$ is a continuous kernel on $\mathcal{X}$. $X'$ (resp. $Y'$) is a random variable independent of $X$ (resp. $Y$). If $k$ is a trivial kernel, MMD degenerates to the energy distance. Mathematically, the "flat" geometry that MMD induces on the space of probability measures does not faithfully lift the ground distance [11], but MMD is cheaper to compute than OT and has a smaller sample complexity, i.e., approximating the distance with samples of measures [13]. We provide the detailed introduction of more distribution divergences in Appendix A.

# 3    Roles of Distribution Divergence and Representation in distributional RL

## 3.1    Distributional RL: From Neural Q-Fitted Iteration to Neural Z-Fitted Iteration

**Neural Q-Fitted Iteration.** It is known that Deep Q Learning [16] can be simplified into *Neural Q-Fitted Iteration* [10] under tricks of experience replay and the target network $Q_{\theta^*}$, where we update parameterized $Q_\theta(s, a)$ in each iteration $k$:

$$Q_\theta^{k+1} = \underset{Q_\theta}{\text{argmin}} \frac{1}{n} \sum_{i=1}^n \left[y_i - Q_\theta^k\left(s_i, a_i\right)\right]^2, \tag{5}$$

where the target $y_i = r(s_i, a_i) + \gamma \max_{a \in \mathcal{A}} Q_{\theta^*}^k\left(s_i', a\right)$ is fixed within every $T_{\text{target}}$ steps to update target network $Q_{\theta^*}$ by letting $\theta^* = \theta$ and the experience buffer induces independent samples $\{(s_i, a_i, r_i, s_i')\}_{i \in [n]}$. In an ideal case that neglects the non-convexity and TD approximation errors, we have $Q_\theta^{k+1} = \mathcal{T} Q_\theta^k$, which is exactly equivalent to updating under Bellman optimality operator.

**Neural Z-Fitted Iteration.** Analogous to neural Q-fitted iteration, we can also simplify value-based distributional RL methods based on a parameterized $Z_\theta$ into a *Neural Z-fitted Iteration* as

$$Z_\theta^{k+1} = \underset{Z_\theta}{\text{argmin}} \frac{1}{n} \sum_{i=1}^n d_p(Y_i, Z_\theta^k\left(s_i, a_i\right)), \tag{6}$$

where the target $Y_i = R(s_i, a_i) + \gamma Z_{\theta^*}^k\left(s_i', \pi_Z(s')\right)$ with $\pi_Z(s') = \text{argmax}_{a'} \mathbb{E}\left[Z_{\theta^*}^k(s', a')\right]$ is fixed within every $T_{\text{target}}$ steps to update target network $Z_{\theta^*}$, and $d_p$ is a divergence metric between two distributions.

## 3.2    Key Roles of $d_p$ and $Z_\theta$

Within the Neural Z-fitted Iteration framework proposed in Eq. 6, we observe that the choice of representation manner on $Z_\theta$ and the metric $d_p$ are pivotal for the distributional RL algorithms. For instance, QR-DQN [8] approximates Wasserstein distance $W_p$, which leverages quantiles to represent

| Algorithm | $d_p$ Distribution Divergence | Representation $Z_\theta$ | Convergence Rate of $\mathfrak{T}^\pi$ | Sample Complexity of $d_p$ |
|---|---|---|---|---|
| C51 [2] | Cramér distance | Histogram | $\sqrt{\gamma}$ | |
| QR-DQN [8] | Wasserstein distance | Quantiles | $\gamma$ | $\mathcal{O}(n^{-\frac{1}{d}})$ |
| MMDDRL [17] | MMD | Samples | $\gamma^{\alpha/2}$ with kernel $k_\alpha$ | $\mathcal{O}(1/n)$ |
| SinkhornDRL (ours) | Sinkhorn divergence | Samples | $\gamma$ $(\varepsilon \to 0)$ 
 $\gamma^{\alpha/2}$ $(\varepsilon \to \infty)$ | $\mathcal{O}(n^{\frac{\kappa}{\varepsilon \lceil d/2 \rceil \sqrt{n}}})$ $(\varepsilon \to 0)$ 
 $\mathcal{O}(n^{-\frac{1}{2}})$ $(\varepsilon \to \infty)$ |

Table 1: Comparison between typical distributional RL algorithms under different distribution divergences and represention of $Z_\theta$. $k_\alpha = -\|x - y\|^\alpha$ in MMDDRL, $d$ is the sample dimension and $\kappa = 2\beta d + \|c\|_\infty$, where the cost function $c$ is $\beta$-Lipschitz [13]. Sample complexity of MMD can be improved to $\mathcal{O}(1/n)$ using kernel herding technique [5].

the distribution of $Z_\theta$. C51 [2] represents $Z_\theta$ via a categorical distribution under the convergence of Cramér distance [3, 19], while MMD distributional RL (MMDDRL) [17] learns samples to represent the distribution of $Z_\theta$ based on MMD. We compare characteristics of these distribution divergence, including the convergence rate and sample complexity, in Table 1. Theoretical results regarding Sinkhorn divergence is based on [13] and the detailed convergence proof of other distances is also provided in Appendix A. In summary, we argue that $d_p$ and $Z_\theta$ are two crucial factors in distributional RL design, based on which we introduce our Sinkhorn distributional RL.

# 4 Sinkhorn Distributional RL (SinkhornDRL)

In this section, we firstly introduce Sinkhorn divergence and apply it in distributional RL. Next, we conduct a theoretical analysis about the convergence speed and a new moment matching manner of our algorithm under the Sinkhorn divergence. Finally, a practical Sinkhorn iteration algorithm is introduced to evaluate the Sinkhorn divergence.

## 4.1 Sinkhorn Divergence and Genetic Algorithm

We design Sinkhorn distributional RL algorithm via Sinkhorn divergence. Sinkhorn divergence [21] is a tractable loss to approximate the optimal transport problem by leveraging an entropic regularization to turn the original Wasserstein distance into a differentiable and more robust quantity. The resulting loss can be computed using Sinkhorn fixed point iterations, which is naturally suitable for modern deep learning frameworks. In particular, the entropic smoothing generates a family of losses interpolating between Wasserstein distance and Maximum Mean Discrepancy (MMD). As such, it allows us to find a sweet trade-off that simultaneously leverages the geometry of Wasserstein distance on the one hand, and the favorable high-dimensional sample complexity and unbiased gradient estimates of MMD. We introduce the entropic regularized Wassertein distance $\mathcal{W}_{c,\varepsilon}(\mu, \nu)$ as

$$\min_{\Pi \in \mathbf{\Pi}(\mu,\nu)} \int c(x,y) \mathrm{d}\Pi(x,y) + \varepsilon \mathrm{KL}(\Pi | \mu \otimes \nu), \tag{7}$$

where $\mathrm{KL}(\Pi | \mu \otimes \nu) = \int \log\left(\frac{\Pi(x,y)}{\mathrm{d}\mu(x)\mathrm{d}\nu(y)}\right) \mathrm{d}\Pi(x,y)$ is a strongly convex regularization. The impact of this entropy regularization is similar to $\ell_2$ ridge regularization in linear regression. Next, the sinkhorn loss [11, 14] between two measures $\mu$ and $\nu$ is defined as

$$\overline{\mathcal{W}}_{c,\varepsilon}(\mu, \nu) = 2\mathcal{W}_{c,\varepsilon}(\mu, \nu) - \mathcal{W}_{c,\varepsilon}(\mu, \mu) - \mathcal{W}_{c,\varepsilon}(\nu, \nu). \tag{8}$$

As demonstrated by [11], the Sinkhorn divergence $\overline{\mathcal{W}}_{c,\varepsilon}(\mu, \nu)$ is convex, smooth and positive definite that metrizes the convergence in law. In statistical physics, $\mathcal{W}_{c,\varepsilon}(\mu, \nu)$ can be re-factored as a projection problem:

$$\mathcal{W}_{c,\varepsilon}(\mu, \nu) := \min_{\Pi \in \mathbf{\Pi}(\mu,\nu)} \mathrm{KL}\left(\Pi | \mathcal{K}\right), \tag{9}$$

where $\mathcal{K}$ is the Gibbs distribution with the density function satisfies $d\mathcal{K}(x,y) = e^{-\frac{c(x,y)}{\varepsilon}} d\mu(x) d\nu(y)$. This problem is often referred to as the "static Schrödinger problem" [15, 20] as it was initially considered in statistical physics.

**Distributional RL with Sinkhorn Divergence and Particle Representation.** The key of applying Sinkhorn divergence in distributional RL is to simply leverage the Sinkhorn loss $\overline{\mathcal{W}}_{c,\varepsilon}$ to measure the distance between the current action-value distribution $Z_\theta(s,a)$ and the target distribution $\mathfrak{T}^\pi Z_\theta(s,a)$, yielding $\overline{\mathcal{W}}_{c,\varepsilon}(Z_\theta(s,a), \mathfrak{T}^\pi Z_\theta(s,a))$ for each $s, a$ pairs. In terms of the representation for $Z_\theta(s,a)$, we employ the unrestricted statistics, i.e., deterministic samples, due to its superiority in MMDDRL [17], instead of using predefined statistic functionals, e.g., quantiles in QR-DQN [8] or histogram partitions in C51 [2]. More concretely, we use neural networks to generate samples that approximate the value distribution. This can be expressed as $Z_\theta(s,a) := \{Z_\theta(s,a)_i\}_{i=1}^N$, where $N$ is the number of generated samples. We refer to the samples $\{Z_\theta(s,a)_i\}_{i=1}^N$ as *particles*. Then we leverage the Dirac mixture $\frac{1}{N}\sum_{i=1}^N \delta_{Z_\theta(s,a)_i}$ to approximate the true density function of $Z^\pi(s,a)$, thus minimizing the Sinkhorn divergence between the approximate distribution and its distributional Bellman target. A detailed and generic distributional RL algorithm with Sinkhorn divergence and particle representation is provided in Algorithm 1.

---

**Algorithm 1** Generic Sinkhorn distributional RL Update

---

**Require**: Number of generated samples $N$, the cost function $c$ and hyperparameter $\varepsilon$.

**Input**: Sample transition $(s, a, r', s')$

  1: **if** Policy evaluation **then**
  2:    $a^* \sim \pi(\cdot|s')$.
  3: **else**
  4:    $a^* \leftarrow \arg\max_{a' \in \mathcal{A}} \frac{1}{N} \sum_{i=1}^{N} Z_\theta (s', a')_i$
  5: **end if**
  6: $\mathfrak{T}Z_i \leftarrow r + \gamma Z_{\theta^*} (s', a^*)_i, \forall 1 \leq i \leq N$

**Output**: $\overline{\mathcal{W}}_{c,\varepsilon} \left( \{Z_\theta(s,a)_i\}_{i=1}^{N}, \{\mathfrak{T}Z_\theta(s,a)_j\}_{j=1}^{N} \right)$

---

**Remark.** By comparing the state-of-the-art MMDDRL algorithm [17], our Sinkhorn distributional RL simply modifies the distribution divergence. Hence, we can also easily extend our generic Sinkhorn algorithm to DQN-like architecture as well as IQN [7] and FQF [24]. A following question is whether there is any theoretical connection between Sinkhorn distributional RL and algorithms based on MMD and Wasserstein distance. We provide this crucial analysis in Section 4.2

## 4.2 Theoretical Analysis under Sinkhorn Divergence

**Convergence Analysis.** Firstly, we denote the supreme form of Sinkhorn divergence as $\overline{\mathcal{W}}_{c,\varepsilon}^{\infty}(\mu, \nu)$:

$$\overline{\mathcal{W}}_{c,\varepsilon}^{\infty}(\mu, \nu) = \sup_{(x,a) \in \mathcal{S} \times \mathcal{A}} \overline{\mathcal{W}}_{c,\varepsilon}(\mu(x,a), \nu(x,a)). \tag{10}$$

We will use $\overline{\mathcal{W}}_{c,\varepsilon}^{\infty}(\mu, \nu)$ to establish the convergence of $\mathfrak{T}^\pi$ in Theorem 1.

**Theorem 1.** *If we leverage Sinkhorn loss $\overline{\mathcal{W}}_{c,\varepsilon}(\mu, \nu)$ in Eq. 8 as the distribution divergence in distributional RL, and **choose the unrectified kernel** $k_\alpha := -\|x - y\|^\alpha$ **as** $-c$ ($\alpha > 0$), it holds that*

*(1) As $\varepsilon \to 0$, $\overline{\mathcal{W}}_{c,\varepsilon}(\mu, \nu) \to 2W_\alpha(\mu, \nu)$. When $\varepsilon = 0$, $\mathfrak{T}^\pi$ is a $\gamma$-contraction under $\overline{\mathcal{W}}_{c,\varepsilon}^{\infty}$.*

*(2) As $\varepsilon \to +\infty$, $\overline{\mathcal{W}}_{c,\varepsilon}(\mu, \nu) \to MMD_{k_\alpha}^2(\mu, \nu)$. When $\varepsilon = +\infty$, $\mathfrak{T}^\pi$ is $\gamma^{\alpha/2}$-contractive under $\overline{\mathcal{W}}_{c,\varepsilon}^{\infty}$.*

*(3) For any $\varepsilon \in (0, +\infty)$, $\mathfrak{T}^\pi$ is a **closely** non-expansive operator under $\overline{\mathcal{W}}_{c,\varepsilon}^{\infty}$, and the difference term $\Delta(\gamma) \to 0$ as $\gamma \to 1$.*

Proof is provided in Appendix B. Theorem 1 (1) and (2) are follow-up conclusions in terms of the convergence behavior of $\mathfrak{T}^\pi$ based on the interpolation relationship between Sinkhorn divergence with Wasserstein distance and MMD [14]. Our key theoretical contribution is for the general $\varepsilon \in (0, \infty)$, the convergence behavior is determined by the "joint" KL divergence in Eq. 9 between the optimal joint distribution $\Pi^*$ and the Gibbs distribution associated with the cost function $c$. We conclude that $\mathfrak{T}^\pi$ is a **close** non-expansive operator and the different term $\Delta(\gamma) \to 0$ as $\gamma \to 1$. Note that $\gamma$ is normally very close to 1 in practice, and this is beneficial for the convergence of $\mathfrak{T}^\pi$ under $\overline{\mathcal{W}}_{c,\varepsilon}^{\infty}$.

**Remark on Theorem 1 (3).** If we consider to use Gaussian kernel, we can not guarantee $\mathfrak{T}^\pi$ is closely non-expansive for any $\varepsilon \in (0, \infty)$. This conclusion is consistent with those discussed in MMDDRL [17], where $\mathfrak{T}^\pi$ is generally not a contraction operator under MMD equipped with Gaussian kernels as a counterexample has been pointed out in MMDDRL (when $\varepsilon \to +\infty$). When $\varepsilon \to 0$, the $\gamma$-contractive $\mathfrak{T}^\pi$ under Wasserstein distance is also not contradictory to Theorem 1 (3). Moreover, although we can only obtain that $\mathfrak{T}^\pi$ is closely non-expansive, the expectation of $Z^\pi$ remains a $\gamma$-contraction (see Appendix B). In experiments, we thereby use $k_\alpha$ and we can also demonstrate the appealing empirical performance of our SinkhornDRL algorithm in Section 5.

**Regularized Moment Matching under Sinkhorn Divergence.** We further examine the potential reason behind the empirical success for SinkhornDRL, although only a non-expansive contraction can be guaranteed for the general case when $\varepsilon \in (0, +\infty)$ as shown in Theorem 1. Inspired by the similar manner in MMDDRL [17], we find that the Sinkhorn divergence with the Gaussian kernel can also promote to match all moments between two distributions. More specifically, the Sinkhorn divergence can be rewritten as a regularized moment matching form in Proposition 1.

**Proposition 1.** *For $\varepsilon \in (0, +\infty)$, Sinkhorn divergence $\overline{\mathcal{W}}_{c,\varepsilon}(\mu, \nu)$ associated with Gaussian kernels $k(x, y) = \exp(-(x - y)^2/(2\sigma^2))$ as $-c$, can be equivalent to*

$$\overline{\mathcal{W}}_{c,\varepsilon}(\mu, \nu) := \sum_{n=0}^{\infty} \frac{1}{\sigma^{2n} n!} \left( \tilde{M}_n(\mu) - \tilde{M}_n(\nu) \right)^2 + \varepsilon \mathbb{E} \left[ \log \frac{(\Pi_\varepsilon^*(X, Y))^2}{\Pi_\varepsilon^*(X, X')\Pi_\varepsilon^*(Y, Y')} \right], \tag{11}$$

*where $\Pi_\varepsilon^*$ denotes the optimal $\Pi$ determined by $\varepsilon$ by evaluating the Sinkhorn divergence via $\min_{\Pi \in \mathbf{\Pi}(\mu,\nu)} \overline{\mathcal{W}}_{c,\varepsilon}(\mu, \nu)$. $\tilde{M}_n(\mu) = \mathbb{E}_{x \sim \mu} \left[ e^{-x^2/(2\sigma^2)} x^n \right]$, and similarly for $\tilde{M}_n(\nu)$.*

We provide the proof of Proposition 1 in Appendix C. Similar to MMDDRL associated with a Gaussian kernel [17], Sinkhorn divergence approximately performs a regularized moment matching scaled by $e^{-x^2/(2\sigma^2)}$. This similar moment matching impact intuitively explains the empirical success of SinkhornDRL as MMDDRL, although the contraction of both MMD with Gaussian kernel [17] and Sinkhorn divergence for general $\epsilon \in (0, +\infty)$ may not be guaranteed.

**Equivalence to Regularized MMD distributional RL.** Based on Proposition 1, we can immediately establish the connection between Sinkhorn divergence and MMD in Corollary 1, indicating that minimizing Sinkhorn divergence between two distributions is equivalent to minimizing a regularized squared MMD.

**Corollary 1.** *For $\varepsilon \in (0, +\infty)$ and denote $\Pi_\varepsilon^*$ as the optimal $\Pi$ by evaluating the Sinkhorn divergence, it holds that*

$$\overline{\mathcal{W}}_{c,\varepsilon} := MMD^2_{-c}(\mu, \nu) + \varepsilon \mathbb{E} \left[ \log \frac{(\Pi_\varepsilon^*(X, Y))^2}{\Pi_\varepsilon^*(X, X')\Pi_\varepsilon^*(Y, Y')} \right], \tag{12}$$

*where we use $\overline{\mathcal{W}}_{c,\varepsilon}$ to replace $\overline{\mathcal{W}}_{c,\varepsilon}(\mu, \nu)$ for short.*

Proof of Corollary 1 is provided in Appendix C. It is worthy of noting that this equivalence is established for the general case when $\varepsilon \in (0, +\infty)$, and it does not hold in the limit cases when $\varepsilon \to 0$ or $+\infty$. For example, when $\varepsilon \to +\infty$, the second part including $\varepsilon$ in Eq. 12 is not expected to dominate. This is owing to the fact that the regularization term would be 0 as $\Pi_\varepsilon^* \to \mu \otimes \nu$ when $\varepsilon \to +\infty$. In summary, even though the Sinkhorn divergence was initially proposed to serve as an entropy regularized Wasserterin distance, it turns out that it is equivalent to a regularized MMD, as revealed in Corollary 1. This connection provides strong evidence for our empirical results, in which SinkhornDRL achieves competitive performance as opposed to MMDDRL.

## 4.3 Distributional RL via Sinkhorn Iterations

The theoretical analysis in Section 4.2 sheds light on the behavior of distributional RL with Sinkhorn divergence, but another crucial issue we need to address is how to evaluate the Sinkhorn loss effectively. Due to the advantages of Sinkhorn divergence that both enjoys geometry property of optimal transport and the computational effectiveness of MMD, we can utilize Sinkhorn's algorithm, i.e., Sinkhorn Iterations [21, 14], to evaluate the Sinkhorn loss. Notably, Sinkhorn iteration with $L$ steps yields a differentiable and solvable efficiently loss function as the main burden involved in it is the matrix-vector multiplication, which streams well on the GPU with simply adding extra differentiable layers on the typical deep neural network, such as a DQN architecture.

Specifically, given two sample sequences $\{Z_i\}_{i=1}^N, \{\mathfrak{T}Z_j\}_{j=1}^N$ in the distributional RL algorithm, the optimal transport distance is equivalent to the form:

$$\min_{P \in \mathbb{R}_+^{N \times N}} \left\{ \langle P, \hat{c} \rangle; P\mathbf{1}_N = \mathbf{1}_N, P^\top \mathbf{1}_N = \mathbf{1}_N \right\}, \tag{13}$$

where the empirical cost function $\hat{c}_{i,j} = c(Z_i, \mathfrak{T}Z_j)$. By adding entropic regularization on optimal transport distance, Sinkhorn divergence can be viewed to restrict the search space of $P$ in the following scaling form:

$$P_{i,j} = a_i \mathcal{K}_{i,j} b_j, \tag{14}$$

where $\mathcal{K}_{i,j} = e^{-\hat{c}_{i,j}/\varepsilon}$ is the Gibbs kernel defined in Eq. 9. This allows us to leverage iterations regarding the vectors $a$ and $b$. More specifically, we initialize $b_0 = \mathbf{1}_N$, and then the Sinkhorn iterations are expressed as

$$a_{l+1} \leftarrow \frac{\mathbf{1}_N}{\mathcal{K} b_l} \quad \text{and} \quad b_{l+1} \leftarrow \frac{\mathbf{1}_N}{\mathcal{K}^\top a_{l+1}}, \tag{15}$$

---

**Algorithm 2** Sinkhorn Iterations to Approximate $\overline{\mathcal{W}}_{c,\varepsilon}\left(\{Z_i\}_{i=1}^N, \{\mathfrak{T}Z_j\}_{j=1}^N\right)$

---

**Input**: Two samples sequences $\{Z_i\}_{i=1}^N, \{\mathfrak{T}Z_j\}_{j=1}^N$, number of Sinkhorn iterations $L$ and hyperparameter $\varepsilon$.

1: $\hat{c}_{i,j} = c(Z_i, \mathfrak{T}Z_j)$ for $\forall i = 1, ..., N, j = 1, ..., N$
2: $\mathcal{K}_{i,j} = \exp(-\hat{c}_{i,j}/\varepsilon)$
3: $b_0 \leftarrow \mathbf{1}_N$
4: **for** $l = 1, 2, ..., L$ **do**
5: $\quad a_l \leftarrow \frac{\mathbf{1}_N}{\mathcal{K}b_{l-1}}, b_l \leftarrow \frac{\mathbf{1}_N}{\mathcal{K}a_l}$
6: **end for**
7: $\widehat{\overline{\mathcal{W}}}_{c,\varepsilon}\left(\{Z_i\}_{i=1}^N, \{\mathfrak{T}Z_j\}_{j=1}^N\right) = \langle (K \odot \hat{c})b, a \rangle$

**Return**: $\widehat{\overline{\mathcal{W}}}_{c,\varepsilon}\left(\{Z_i\}_{i=1}^N, \{\mathfrak{T}Z_j\}_{j=1}^N\right)$

---

where $\dot{-}$ indicates an entry-wise division. It has been proven that Sinkhorn iteration asymptotically converges to the true loss in a linear rate [14, 12, 6]. We provide a detailed algorithm description of Sinkhorn iterations in Algorithm 2. With the efficient and differential Sinkhorn iterations, we can easily evaluate the Sinkhorn divergence and thus let our algorithm enjoy its theoretical advantages. In practice, we need to choose $L$ and $\varepsilon$, and we conduct a rigorous sensitivity analysis in Section 5.

## 5 Experiments

We demonstrate the effectiveness of SinkhornDRL as described in Algorithm 1 on the full 55 Atari 2600 games. Specifically, we leverage the same architecture as QR-DQN [8], and replace the quantiles output with $N$ particles, i.e., samples. In contrast to MMDDRL, SinkhornDRL only changes the distribution divergence from MMD to Sinkhorn divergence, and therefore the potential superiority in the performance can be attributed to the advantages of Sinkhorn divergence. In Section 5.1, we make a rigorous comparison between SinkhornDRL with other typical distributional RL algorithms from the perspectives of learning curves and final ratio improvement of returns. An extensive sensitivity analysis in terms of multiple hyperparameters in SinkhornDRL is provided in Section 5.2.

**Baselines.** Due to the interpolation characteristic of Sinkhorn divergence between Wassertein distance and MMDDRL, we choose three typical distributional RL algorithms as classic baselines, including QR-DQN [8] that approximates the Wasserstein distance, C51 [2] and MMDDRL [17], as well as DQN [16]. MMDDRL algorithm is implemented with the same architecture as QRDQN, and leverages Gaussian kernels $k_h(x, y) = \exp(-(x-y)^2/h)$ with the kernel mixture trick covering a range of bandwidths $h$, which is same as the basic setting in the original MMDDQN paper [17]. We deploy all algorithms on 55 Atari 2600 games, and reported results are averaged over 3 seeds with the shade indicating the standard deviation.

**Hyperparameter settings.** For a fair comparison with QR-DQN, C51 and MMDDRL, we used the same hyperparamters: the number of generated samples $N = 200$, Adam optimizer with lr $= 0.00005, \epsilon_{\text{Adam}} = 0.01/32$. We used a target network to compute the distributional Bellman target, which fits well in the neural Z-fitted iteration framework. In addition, we choose number of Sinkhorn iterations $L = 10$ and smoothing hyperparameter $\varepsilon = 10.0$ in Section 5.1 as they are not sensitive within a proper interval as demonstrated in Section 5.2. We choose the unrectified kernel as the cost function, i.e., $-c = k_\alpha$, and select $\alpha = 2$ in $k_\alpha$ in our SinkhornDRL algorithm.

### 5.1 Performance of SinkhornDRL

Figure 1 illustrates that SinkhornDRL can achieve the competitive performance across 55 Atari games compared with various baseline algorithms with different metrics $d_p$ and representation manners on $Z_\theta$. On a large number of games, e.g., Tennis, Seaquest and Atlantis, SinkhornDRL can significantly outperform other baselines, especially on Tennis where other algorithms even fail to converge. The improvement of SinkhornDRL over MMDDRL empirically verifies the regularization advantage of the Sinkhorn as analyzed in Corollary 1. On some games, e.g., Breakout, Pong and SpaceInvaders,

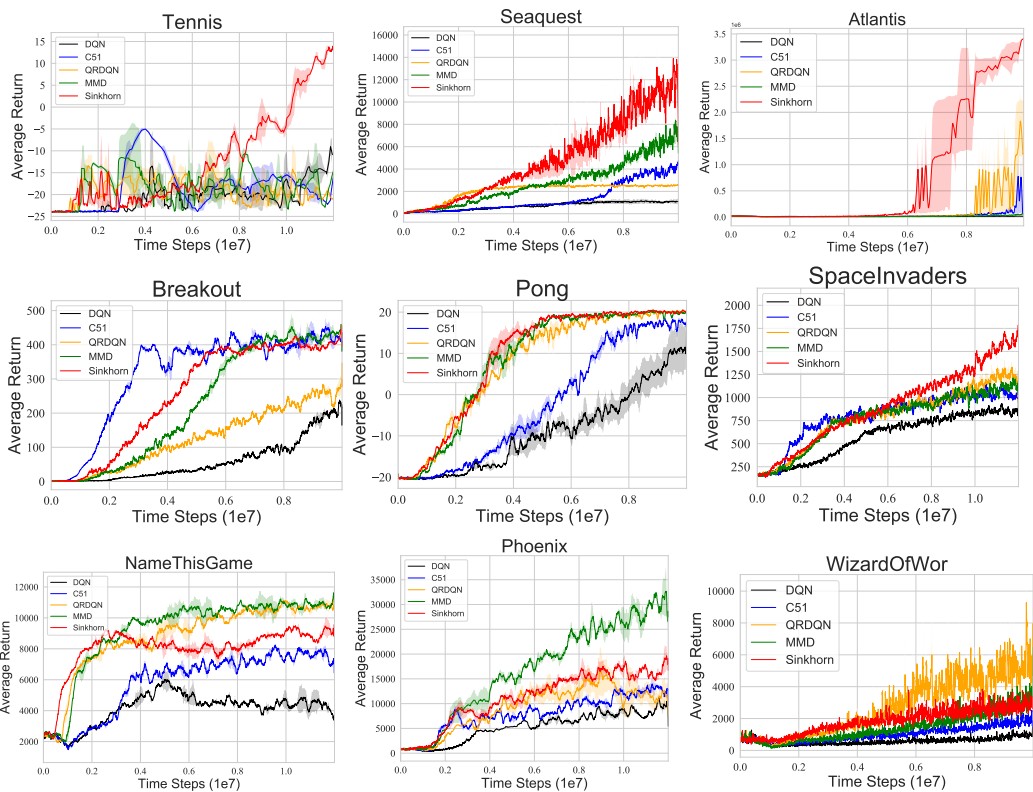

Figure 1: Learning curves of SinkhornDRL algorithm compared with DQN, C51, QR-DQN and MMD, on nine typical Atari games over 3 seeds.

SinkhornDRL is on par with MMDDRL and other baselines, while on the last row in Figure 1, SinkhornDRL is slightly inferior to the state-of-the-art algorithm. We provide learning curves of all typical distributional RL algorithms on all 55 Atari games in Appendix E, where SinkhornDRL still achieves the competitive performance in general.

To further demonstrate theoretical properties of SinkhornDRL in Theorem 1, we conduct a ratio improvement comparison across 55 Atari games between SinkhornDRL with QRDQN and MMDDRL, respectively. Figure 2 showcases that by comparing with QRDQN (left), SinkhornDRL achieves better performance across more than half of considered games. More importantly, the superiority of SinkhornDRL is significant across a large amount of games, including Venture, Seaquest, Tennis and Phoenix. This empirical outperformance verifies the effectiveness and potential of smoothing Wassertein distance in distributional RL, e.g., Sinkhorn divergence. In contrast with MMDDRL, the superiority of SinkhornDRL is reduced with the performance improvement only on a small proportion of games, while a remarkable boost of performance for SinkhornDRL on a large amount of games can be easily observed. We also report mean and median of best human-normalized scores in Table 2 of Appendix D, where SinkhornDRL achieves almost state-of-the-art performance as MMDDRL on average.

Therefore, we conclude that SinkhornDRL is competitive with the state-of-the-art distributional RL algorithms, e.g., MMDDRL, and can be extremely superior over existing algorithms on a large proportion of games. This empirical success can be owing to theoretical advantage of Sinkhorn divergence that simultaneously makes full use of the data geometry from Wasserstein distance and the unbiased gradient estimate property from MMD, which coincides with results in Theorem 1.

## 5.2 Sensitivity Analysis and Computational Cost

Figure 3 (a) suggests the performance of our algorithm is robust to $\varepsilon$ in a certain range, e.g., $[1, 500]$, facilitating its deployment in practice. If we increase $\varepsilon$, SinkhornDRL's performance tends to MMD,

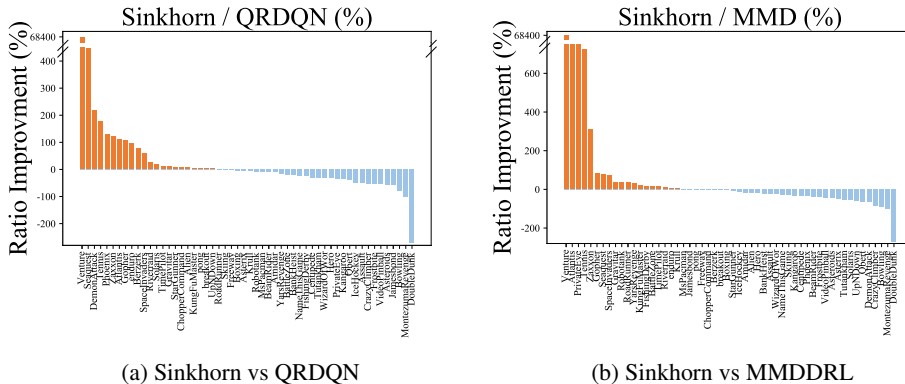

(a) Sinkhorn vs QRDQN        (b) Sinkhorn vs MMDDRL

Figure 2: Ratio improvement of return for Sinkhorn distributional RL algorithm over QRDQN (left) and MMDDRL (right) over 3 seeds. For example, the ratio improvement is calculated by (Sinkhorn - QRDQN) / QRDQN in the left.

while if we gradually decline $\varepsilon$, SinkhornDRL's performance tends to QR-DQN. It is also noted that Sinkhorn iterations in Algorithm 2 will suffer from the numerical instability issue under an overly small or large $\varepsilon$. More results with the discussion are provided in Appendix F. It is also illustrated that our algorithm is insensitive to the number of iterations $L$ and samples $N$ as well, but an overly large $N$ can slightly worsen the performance of SinkhornDRL, and at the same time increases the computational burden. Therefore, a proper number of samples, e.g., 200, is sufficient to attain an appealing performance with the computational effectiveness.

For the computation cost, SinkhronDRL indeed increases around 50% computation cost compared with QR-DQN and C51, but only slightly increases the cost (by around 20%) in contrast to MMDDRL. Detailed comparison is given in Appendix F.

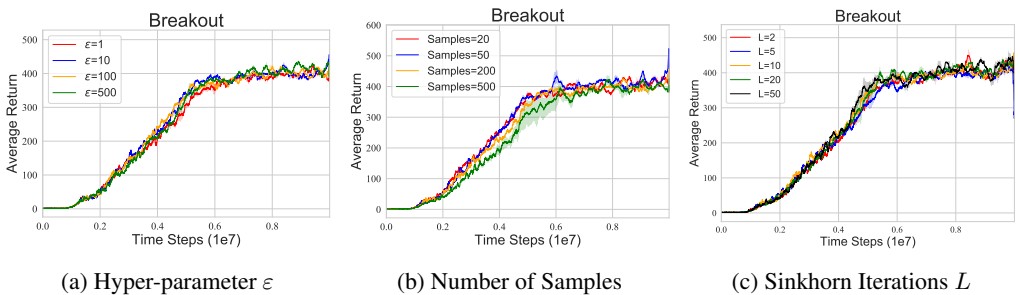

(a) Hyper-parameter $\varepsilon$      (b) Number of Samples      (c) Sinkhorn Iterations $L$

Figure 3: Sensitivity analysis of SinkhornDRL on Breakout regarding $\varepsilon$, number of samples, and number of iteration $L$. Learning curves are reported over 3 seeds.

## 6 Discussions and Conclusion

The main limitation of our proposal is that the superiority over existing state-of-the-art algorithms may not be sufficiently significant. To extend our algorithm for better performance, implicit generative models, including parameterizing the cost function in Sinkhorn loss, can be further incorporated. We leave it as the future work. Moreover, other divergences, e.g., thoses that can also smooth Wassertein distance, can also be applied into the design of distributional RL algorithms in the future.

In this paper, a novel family of distributional RL algorithms based on Sinkhorn Divergence is proposed that accomplishes a competitive performance compared with the-state-of-the-art distributional RL algorithms on 55 Atari games. Theoretical analysis about the convergence and moment matching behavior is provided along with a rigorous empirical verification. Albeit being associated with MMD algorithm, distributional RL with Sinkhorn divergence is complementary to previous algorithms, leading to an important contribution among the research community.

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
