with the cumulative distribution function as $F$. Further, $\ell_p$ distance [9] is defined as

$$\ell_p(X,Y) := \left( \int_{-\infty}^{\infty} |F_X(\omega) - F_Y(\omega)|^p \, d\omega \right)^{1/p} = \| F_X - F_Y \|_p \tag{17}$$

The $\ell_p$ distance and Wassertein metric are identical at $p = 1$, but are otherwise distinct. Note that when $p = 2$, $\ell_p$ distance is also called Cramér distance [3] $d_C(X,Y)$. Also, the Cramér distance has a different representation given by

$$d_C(X,Y) = \mathbb{E}|X - Y| - \frac{1}{2}\mathbb{E}\,|X - X'| - \frac{1}{2}\mathbb{E}\,|Y - Y'|\,, \tag{18}$$

where $X'$ and $Y'$ are the i.i.d. copies of $X$ and $Y$. Energy distance [22, 27] is a natural extension of Cramér distance to the multivariate case, which is defined as

$$d_E(\mathbf{X},\mathbf{Y}) = \mathbb{E}\|\mathbf{X} - \mathbf{Y}\| - \frac{1}{2}\mathbb{E}\|\mathbf{X} - \mathbf{X}'\| - \frac{1}{2}\mathbb{E}\|\mathbf{Y} - \mathbf{Y}'\|, \tag{19}$$

where $\mathbf{X}$ and $\mathbf{Y}$ are multivariate. Moreover, the energy distance is a special case of the maximum mean discrepancy (MMD), which is formulated as

$$\text{MMD}(\mathbf{X},\mathbf{Y};k) = \left( \mathbb{E}\left[ k\left(\mathbf{X},\mathbf{X}'\right)\right] + \mathbb{E}\left[ k\left(\mathbf{Y},\mathbf{Y}'\right)\right] - 2\mathbb{E}[k(\mathbf{X},\mathbf{Y})] \right)^{1/2} \tag{20}$$

where $k(\cdot,\cdot)$ is a continuous kernel on $\mathcal{X}$. In particular, if $k$ is a trivial kernel, MMD degenerates to energy distance. Additionally, we further define the supreme MMD, which is a functional $\mathcal{P}(\mathcal{X})^{\mathcal{S}\times\mathcal{A}} \times \mathcal{P}(\mathcal{X})^{\mathcal{S}\times\mathcal{A}} \to \mathbb{R}$ defined as

$$\text{MMD}_\infty(\mu,\nu) = \sup_{(x,a)\in\mathcal{S}\times\mathcal{A}} \text{MMD}_\infty(\mu(x,a),\nu(x,a)) \tag{21}$$

We further present the convergence rate under different distribution divergences.

- $\mathcal{T}^\pi$ is $\gamma$-contractive under the supreme form of Wassertein distance $W_p$.

- $\mathcal{T}^\pi$ is $\gamma^{1/p}$-contractive under the supreme form of $\ell_p$ distance.

- $\mathcal{T}^\pi$ is $\gamma^{\alpha/2}$-contractive under $\text{MMD}_\infty$ with the kernel $k_\alpha(x,y) = -\|x-y\|^\alpha, \forall \alpha > 0$.

**Proof of Contraction.**

- Contraction under supreme form of Wasserstein diatance is provided in Lemma 3 [2].

- Contraction under supreme form of $\ell_p$ distance can refer to Theorem 3.4 [9].

- Contraction under $\text{MMD}_\infty$ is provided in Lemma 6 [17].

# B Proof of Theorem 1

*Proof.* **1.** As $\varepsilon \to 0$ and $c = -k_\alpha$, it is obvious to observe that Sinkhorn loss degenerates to the wasserstein distance. We also have the conclusion that the distributional Bellman operator $\mathfrak{T}^\pi$ is $\gamma$-contractive under the supreme form of Wasserstein diatance, the proof of which is provided in Lemma 3 [2]. Since the above conclusion is made directly based on the limiting case when $\varepsilon = 0$, for an unspecified $\varepsilon$, we need a more rigorous proof. We show that their distance difference is **at most an infinitesimal** $\delta$.

Firstly, as $\mathcal{W}_{c,\varepsilon} \to W_\alpha$ and the regularization term is non-negative, using the language of $(\varepsilon,\delta)$ definition, we have: for $\forall \delta$, there exists a small positive constant $a$, such that $\mathcal{W}_{c,\varepsilon} - W_\alpha < \delta$ when $\epsilon \le a$. Based on that, we have the contraction conclusion:

$$
\begin{aligned}
\overline{\mathcal{W}}^\infty_{-\kappa_\alpha,\varepsilon}(\mathfrak{T}^\pi Z_1, \mathfrak{T}^\pi Z_2) &= \overline{\mathcal{W}}^\infty_{-\kappa_\alpha,\varepsilon}(\mathfrak{T}^\pi Z_1, \mathfrak{T}^\pi Z_2) - W^\infty_\alpha(\mathfrak{T}^\pi Z_1, \mathfrak{T}^\pi Z_2) + W^\infty_\alpha(\mathfrak{T}^\pi Z_1, \mathfrak{T}^\pi Z_2) \\
&\le \delta + W^\infty_\alpha(\mathfrak{T}^\pi Z_1, \mathfrak{T}^\pi Z_2),
\end{aligned}
\tag{22}
$$

where the second term $W^\infty_\alpha(\mathfrak{T}^\pi Z_1, \mathfrak{T}^\pi Z_2)$ is contractive, and thus for the unspecified $\varepsilon$, the only difference from the limting $\varepsilon = 0$ is an infinitesimal $\delta$, which will vanish as $\varepsilon \to 0$ or $(a \to 0)$.

**2.** As $\varepsilon \to \infty$, our complete proof is inspired by [18, 14]. Recap the definition of squared MMD is

$$
\mathbb{E}\left[k\left(\mathbf{X},\mathbf{X}'\right)\right] + \mathbb{E}\left[k\left(\mathbf{Y},\mathbf{Y}'\right)\right] - 2\mathbb{E}[k(\mathbf{X},\mathbf{Y})]
$$

When the kernel function $k$ degenerates to a unrectified $k_\alpha(x,y) := -\|x-y\|^\alpha$ for $\alpha \in (0,2)$, the squared MMD would degenerate to

$$
\mathbb{E}\|\mathbf{X}-\mathbf{X}'\|^\alpha + \mathbb{E}\|\mathbf{Y}-\mathbf{Y}'\|^\alpha - 2\mathbb{E}\|\mathbf{X}-\mathbf{Y}\|^\alpha
$$

On the other hand, we have the Sinkhorn loss as

$$
\overline{\mathcal{W}}_{c,\infty}(\mu,\nu) = 2\mathcal{W}_{c,\infty}(\mu,\nu) - \mathcal{W}_{c,\infty}(\nu,\nu) - \mathcal{W}_{c,\infty}(\mu,\mu)
$$

Denoting $\Pi_\varepsilon$ be the unique minimizer for $\overline{\mathcal{W}}_{c,\varepsilon}$, it holds that $\Pi_\varepsilon \to \mu \otimes \nu$ as $\varepsilon \to \infty$. That being said, $\mathcal{W}_{c,\infty}(\mu,\nu) \to \int c(x,y)\mathrm{d}\mu(x)\mathrm{d}\nu(y) + 0 = \int c(x,y)\mathrm{d}\mu(x)\mathrm{d}\nu(y)$. If $c = -k_\alpha = \|x-y\|^\alpha$, we eventually have $\mathcal{W}_{-k_\alpha,\infty}(\mu,\nu) \to \int \|x-y\|^\alpha \mathrm{d}\mu(x)\mathrm{d}\nu(y) = \mathbb{E}\|\mathbf{X}-\mathbf{Y}\|^\alpha$. Finally, we can have

$$
\overline{\mathcal{W}}_{-k_\alpha,\infty} \to 2\mathbb{E}\|\mathbf{X}-\mathbf{Y}\|^\alpha - \mathbb{E}\|\mathbf{X}-\mathbf{X}'\|^\alpha - \mathbb{E}\|\mathbf{Y}-\mathbf{Y}'\|^\alpha
$$

which is exactly the form of squared MMD. Now the key is prove that $\Pi_\varepsilon \to \mu \otimes \nu$ as $\varepsilon \to \infty$.

Firstly, it is apparent that $\mathcal{W}_{c,\varepsilon}(\mu,\nu) \le \int c(x,y)\mathrm{d}\mu(x)\mathrm{d}\nu(y)$ as $\mu \otimes \nu \in \Pi(\mu,\nu)$. Let $\{\varepsilon_k\}$ be a positive sequence that diverges to $\infty$, and $\Pi_k$ be the corresponding sequence of unique minimizers for $\mathcal{W}_{c,\varepsilon}$. According to the optimality condition, it must be the case that $\int c(x,y)\mathrm{d}\Pi_k + \varepsilon_k \mathrm{KL}(\Pi_k, \mu \otimes \nu) \le \int c(x,y)\mathrm{d}\mu \otimes \nu + 0$ (when $\Pi(\mu,\nu) = \mu \otimes \nu$). Thus,

$$
\mathrm{KL}\left(\Pi_k, \mu \otimes \nu\right) \le \frac{1}{\varepsilon_k}\left(\int c\,\mathrm{d}\mu \otimes \nu - \int c\,\mathrm{d}\Pi_k\right) \to 0.
$$

Besides, by the compactness of $\Pi(\mu,\nu)$, we can extract a converging subsequence $\Pi_{n_k} \to \Pi_\infty$. Since KL is weakly lower-semicontinuous, it holds that

$$
\mathrm{KL}\left(\Pi_\infty, \mu \otimes \nu\right) \le \lim_{k \to \infty}\inf \mathrm{KL}\left(\Pi_{n_k}, \mu \otimes \nu\right) = 0
$$

Hence $\Pi_\infty = \mu \otimes \nu$. That being said that the optimal coupling is simply the product of the marginals, indicating that $\Pi_\varepsilon \to \mu \otimes \nu$ as $\varepsilon \to \infty$. As a special case, when $\alpha = 1$, $\overline{\mathcal{W}}_{-k_1,\infty}(u,v)$ is equivalent to the energy distance

$$
d_E(\mathbf{X},\mathbf{Y}) := 2\mathbb{E}\|\mathbf{X}-\mathbf{Y}\| - \mathbb{E}\|\mathbf{X}-\mathbf{X}'\| - \mathbb{E}\|\mathbf{Y}-\mathbf{Y}'\|.
\tag{23}
$$

In summary, if the cost function is the rectified kernel $k_\alpha$, it is the case that $\overline{\mathcal{W}}_{-k_\alpha,\varepsilon}$ converges to the squared MMD as $\varepsilon \to \infty$. According to [17], $\mathfrak{T}^\pi$ is $\gamma^{\alpha/2}$-contractive in the supreme form of MMD with the rectified kernel $k_\alpha$.

For the unspecified $\varepsilon$, we can get the similar result to the case of $\varepsilon \to 0$. For $\forall \delta$, there exists a large positive constant $M$, such that $\mathrm{MMD}^2_{k_\alpha} - \mathcal{W}_{c,\varepsilon} < \delta$ when $\epsilon \ge M$. Based on that, we have the contraction conclusion:

$$
\begin{aligned}
\overline{\mathcal{W}}^\infty_{-\kappa_\alpha,\varepsilon}(\mathfrak{T}^\pi Z_1, \mathfrak{T}^\pi Z_2) &= \overline{\mathcal{W}}^\infty_{-\kappa_\alpha,\varepsilon}(\mathfrak{T}^\pi Z_1, \mathfrak{T}^\pi Z_2) - \mathrm{MMD}^2_\infty(\mathfrak{T}^\pi Z_1, \mathfrak{T}^\pi Z_2) + \mathrm{MMD}^2_\infty(\mathfrak{T}^\pi Z_1, \mathfrak{T}^\pi Z_2) \\
&\le \mathrm{MMD}^2_\infty(\mathfrak{T}^\pi Z_1, \mathfrak{T}^\pi Z_2) - \delta,
\end{aligned}
\tag{24}
$$

where the first term $\mathrm{MMD}^2_\infty(\mathfrak{T}^\pi Z_1, \mathfrak{T}^\pi Z_2)$ is $\gamma^{\frac{\alpha}{2}}$}-contractive, and thus for the unspecified $\varepsilon$, the only difference from the limiting $\varepsilon = \infty$ is an infinitesimal $\delta$, which will vanish as $\varepsilon \to +\infty$ or $(M \to +\infty)$.

471 **3.** For $\varepsilon \in (0, +\infty)$, a key observation for the analysis is that the Sinkhorn divergence would
472 degenerate to a two-dimensional KL divergence, and therefore embraces a similar convergence
473 behavior to KL divergence. Concretely, according to the equivalent form of $\mathcal{W}_{c,\varepsilon}(\mu, \nu)$ in Eq. 9, it
474 can be expressed as the KL divergence between an optimal joint distribution and a Gibbs distribution
475 associated with the cost function:

$$\mathcal{W}_{c,\varepsilon}(\mu, \nu) := \mathrm{KL}\left(\Pi^*(\mu, \nu) | \mathcal{K}(\mu, \nu)\right), \tag{25}$$

476 where $\Pi^*$ is the optimal joint distribution. Thus, the total Sinkhorn divergence is expressed as

$$\overline{\mathcal{W}}_{c,\varepsilon}(\mu, \nu) := 2\mathrm{KL}\left(\Pi^*(\mu, \nu) | \mathcal{K}(\mu, \nu)\right) - \mathrm{KL}\left(\Pi^*(\mu, \mu) | \mathcal{K}(\mu, \mu)\right) - \mathrm{KL}\left(\Pi^*(\nu, \nu) | \mathcal{K}(\nu, \nu)\right). \tag{26}$$

477 Due to the form of $\overline{\mathcal{W}}_{c,\varepsilon}(\mu, \nu)$, the convergence behavior is determined by $\mathcal{W}_{c,\varepsilon}(\mu, \nu)$, which is
478 similar to the behavior of KL divergence. Thus, we will focus on the convergence analysis of
479 $\mathcal{W}_{c,\varepsilon}(\mu, \nu)$. We firstly elaborate a Lemma regarding to the convergence under KL divergence.

480 **Lemma 1.** *Denote the supreme of $D_{KL}$ as $D_{KL}^\infty$, we have: (1) $\mathfrak{T}^\pi$ is a non-expansive operator under*
481 *$D_{KL}^\infty$, i.e., $D_{KL}^\infty(\mathfrak{T}^\pi Z_1, \mathfrak{T}^\pi Z_2) \leq D_{KL}^\infty(Z_1, Z_2)$, (2) the expectation of $Z^\pi$ is still $\gamma$-contractive under*
482 *$D_{KL}^\infty$, i.e., $\|\mathbb{E}\mathfrak{T}^\pi Z_1 - \mathbb{E}\mathfrak{T}^\pi Z_2\|_\infty \leq \gamma \|\mathbb{E}Z_1 - \mathbb{E}Z_2\|_\infty$.*

483 *Proof.* (1) We recap three crucial properties of a divergence metric. The first is *scale sensitive* (**S**)
484 (of order $\beta$, $\beta > 0$), i.e., $d_p(cX, cY) \leq |c|^\beta d_p(X, Y)$. The second property is *shift invariant* (**I**),
485 i.e., $d_p(A + X, A + Y) \leq d_p(X, Y)$. The last one is *unbiased gradient* (**U**). We use $p$ and $q$ to
486 denote the density function of two random variables $X$ and $Y$, and thus $D_{KL}(X, Y)$ is defined as
487 $D_{KL}(X, Y) = \int_{-\infty}^{\infty} p(x) \frac{p(x)}{q(x)} \, \mathrm{d}x$. Firstly, we show that $D_{KL}(X, Y)$ is NOT scale sensitive:

$$\begin{aligned} D_{\mathrm{KL}}(aX, aY) &= \int_{-\infty}^{\infty} \frac{1}{a} p(\frac{x}{a}) \log \frac{\frac{1}{a}p(\frac{x}{a})}{\frac{1}{a}q(\frac{x}{a})} \, \mathrm{d}x \\ &= \int_{-\infty}^{\infty} p(y) \log \frac{p(y)}{q(y)} \, \mathrm{d}y \\ &= D_{\mathrm{KL}}(X, Y), \text{ with } \beta = 0 \end{aligned} \tag{27}$$

488 We further show that $D_{\mathrm{KL}}(X, Y)$ is shift invariant:

$$\begin{aligned} D_{\mathrm{KL}}(A + X, A + Y) &= \int_{-\infty}^{\infty} p(x - A) \log \frac{p(x - A)}{q(x - A)} \, \mathrm{d}x \\ &= \int_{-\infty}^{\infty} p(y) \log \frac{p(y)}{q(y)} \, \mathrm{d}y \\ &= D_{\mathrm{KL}}(X, Y) \end{aligned} \tag{28}$$

489 Moreover, it is well-known that KL divergence has unbiased sample gradients [3]. The supreme $D_{\mathrm{KL}}$
490 is a functional $\mathcal{P}(\mathcal{X})^{\mathcal{S} \times \mathcal{A}} \times \mathcal{P}(\mathcal{X})^{\mathcal{S} \times \mathcal{A}} \to \mathbb{R}$ defined as

$$D_{\mathrm{KL}}^\infty(\mu, \nu) = \sup_{(x,a) \in \mathcal{S} \times \mathcal{A}} D_{\mathrm{KL}}(\mu(x, a), \nu(x, a)) \tag{29}$$

491 Therefore, we prove $\mathfrak{T}^\pi$ is at best a non-expansive operator under the supreme form of $D_{\mathrm{KL}}$:

$$\begin{aligned} &D_{\mathrm{KL}}^\infty(\mathfrak{T}^\pi Z_1, \mathfrak{T}^\pi Z_2) \\ &= \sup_{s,a} D_{\mathrm{KL}}(\mathfrak{T}^\pi Z_1(s, a), \mathfrak{T}^\pi Z_2(s, a)) \\ &= \sup_{s,a} D_{\mathrm{KL}}(R(s, a) + \gamma Z_1(S', A'), R(s, a) + \gamma Z_2(S', A')) \\ &= D_{\mathrm{KL}}(Z_1(S', A'), Z_2(S', A')) \\ &\leq \sup_{s',a'} D_{\mathrm{KL}}(Z_1(s', a'), Z_2(s', a')) \\ &= D_{\mathrm{KL}}^\infty(Z_1, Z_2) \end{aligned} \tag{30}$$

492 There we have $D_{\mathrm{KL}}^{\infty}(\mathfrak{T}^{\pi} Z_1, \mathfrak{T}^{\pi} Z_2) \leq D_{\mathrm{KL}}^{\infty}(Z_1, Z_2)$, implying that $\mathfrak{T}^{\pi}$ is a non-expansive operator
493 under $D_{\mathrm{KL}}^{\infty}$.

494 (2) This statement is an immediate conclusion based on the Lemma 4 in [2]. We give the proof for
495 the completeness. This conclusion holds because the $\mathfrak{T}^{\pi}$ degenerates to $\mathcal{T}^{\pi}$ regardless of the metric
496 $d_p$. Specifically, due to the linearity of expectation, we obtain that

$$\|\mathbb{E}\mathfrak{T}^{\pi} Z_1 - \mathbb{E}\mathfrak{T}^{\pi} Z_2\|_{\infty} = \|\mathcal{T}^{\pi}\mathbb{E}Z_1 - \mathcal{T}^{\pi}\mathbb{E}Z_2\|_{\infty} \leq \gamma\|\mathbb{E}Z_1 - \mathbb{E}Z_2\|_{\infty}. \tag{31}$$

497 This implies that the expectation of $Z$ under $D_{\mathrm{KL}}$ exponentially converges to the expectation of $Z^*$,
498 i.e., $\gamma$-contraction.

499 $\square$

500 We show that $\mathcal{W}_{c,\varepsilon}(\mu, \nu)$ is NOT scale sensitive. Firstly, we denote $\Pi^2$ as the optimal joint distribution
501 for $(U, V)$ and thus we write the explicit form of Sinkhorn divergence $W_{c,\varepsilon}(U, V)$ between two
502 random variables $U$ and $V$:

503 ***

$$W_{c,\varepsilon}(U, V) = \mathrm{KL}(\Pi^2 \| \mathcal{K}) \tag{32}$$

$$= \int_{-\infty}^{\infty} \int_{-\infty}^{\infty} \Pi^2(x, y) \log \frac{\Pi^2(x, y)}{\frac{1}{Z_2} e^{-\frac{c(x,y)}{\epsilon}} \mu(x)\nu(y)} dx dy, \tag{33}$$

504 ***

505 where the normalization factor $Z_2$ for the Gibbs kernel $\mathcal{K}$ is $Z_2 = \int_{-\infty}^{\infty} \int_{-\infty}^{\infty} e^{-\frac{c(x,y)}{\epsilon}} \mu(x)\nu(y) dx dy$
506 and $\mu(x), \nu(y)$ are the marginal density funtion of $U$ and $V$ with respect to $x$ and $y$. We
507 also denote $\Pi^1$ as the optimal joint distribution for $(aU, aV)$. A key proof element is about
508 the Gibbs kernel $\mathcal{K}$. By definition, the pdf of $\mathcal{K}(U, V) \propto e^{\frac{-c(x,y)}{\varepsilon}} \mu(x)\nu(y)$. After a scal-
509 ing transformation, the pdf of $aU$ and $aV$ with respect to $x$ and $y$ would be $\frac{1}{a}\mu(\frac{x}{a})$ and
510 $\frac{1}{a}\nu(\frac{y}{a})$. Thus $\mathcal{K}(2U, 2V) \propto e^{\frac{-c(x,y)}{\varepsilon}} \frac{1}{a}\mu(\frac{x}{a}) \frac{1}{a}\nu(\frac{y}{a})$. The new normalization factor $Z_1$ is $Z_1 =$
511 $\int_{-\infty}^{\infty} \int_{-\infty}^{\infty} \frac{1}{a^2} e^{-\frac{c(x',y')}{\epsilon}} \mu(x'/a)\nu(y'/a) dx' dy' = \int_{-\infty}^{\infty} \int_{-\infty}^{\infty} e^{-\frac{c(ax,ay)}{\epsilon}} \mu(x)\nu(y) dx dy$, the cost func-
512 tion of which is different from $Z_2$. For $\Pi^2(U, V)$, the scaled pdf of $\Pi^2(aU, aV)$ would be $\frac{1}{a^2}\Pi^2(\frac{x}{a}, \frac{y}{a})$.
513 Then we have the following results:

514 ***

$$\mathcal{W}_{c,\varepsilon}(aU, aV) = \mathrm{KL}(\Pi^1 \| \mathcal{K}) \tag{34}$$

$$\leq \mathrm{KL}(\Pi^2 \| \mathcal{K}) \tag{35}$$

$$= \int_{-\infty}^{\infty} \int_{-\infty}^{\infty} \frac{1}{a^2}\Pi^2(\frac{x'}{a}, \frac{y'}{a}) \log \frac{\frac{1}{a^2}\Pi^2(\frac{x'}{a}, \frac{y'}{a})}{\frac{1}{a^2}\frac{1}{Z_1} e^{-\frac{c(x',y')}{\epsilon}} \mu(\frac{x'}{a})\nu(\frac{y'}{a})} dx' dy', \tag{36}$$

$$= \int_{-\infty}^{\infty} \int_{-\infty}^{\infty} \Pi^2(x, y) \log \frac{\Pi^2(x, y)}{\frac{1}{Z_1} e^{-\frac{c(ax,ay)}{\epsilon}} \mu(x)\nu(y)\frac{Z_2}{Z_2}} dx dy, \tag{37}$$

$$= \int_{-\infty}^{\infty} \int_{-\infty}^{\infty} \Pi^2(x, y)(\log \frac{\Pi^2(x, y)}{\frac{1}{Z_1} e^{-\frac{c(ax,ay)}{\epsilon}} \mu(x)\nu(y)} + \log \frac{Z_1}{Z_2}) dx dy, \tag{38}$$

$$\overset{c=-k_\alpha, a\leq 1}{\leq} \int_{-\infty}^{\infty} \int_{-\infty}^{\infty} \Pi^2(x, y) \log \frac{\Pi^2(x, y)}{\frac{1}{Z_1} e^{-\frac{c(x,y)}{\epsilon}} \mu(x)\nu(y)} dx dy + \log \frac{Z_1}{Z_2} \cdot 1, \tag{39}$$

$$= \mathcal{W}_{c,\varepsilon}(U, V) + \Delta_{\mu,\nu}^c(a), \tag{40}$$

515 ***

516 where the second positive term $\Delta_{\mu,\nu}^c(a) = \log \frac{Z_1}{Z_2}$ satisfies $\Delta_{\mu,\nu}^c(a) \to 0$ as $a \to 1$ (in practice $\gamma$
517 is very close to 1). The second inequality holds for the general $\varepsilon$ because for the unrectified kernel
518 $k_\alpha = -\|x - y\|^\alpha$ **with** $a \leq 1$, for any $\varepsilon$ and $x, y$ we have

$$(ax-ay)^\alpha \leq |a|^\alpha(x-y)^\alpha \leq (x-y)^\alpha e^{-\frac{c(ax,ay)}{\varepsilon}} \geq e^{-\frac{c(x,y)}{\varepsilon}} \Rightarrow e^{-\frac{c(ax,ay)}{\varepsilon}} \mu(x)\nu(y) \geq e^{-\frac{c(x,y)}{\varepsilon}} \mu(x)\nu(y)$$

However, under this condition, $Z_1 \geq Z_2$ and thus $\Delta^c_{\mu,\nu}(a) \geq 0$, but $\Delta^c_{\mu,\nu}(a) \to 0$ as $a \to 1$ (in practice $\gamma$ is very close to 1). We think there is indeed a gap between a (close) non-expansion property of Sinkhorn divergence and the empirical success of SinkhornDRL algorithm. The inequality is established based on the unrectified kernel, but it is tricky to find the contrative property for Sinkhorn divergence with the Gaussian kernel for any $\varepsilon$ and $x, y$. Thus, it is fair that some counterexamples may exist for the non-contractive $\mathfrak{T}^\pi$ under Sinkhorn divergence, which is also consistent with the counterexample MMD with Gaussian kernel (when $\varepsilon \to \infty$).

Now we show that $\mathcal{W}_{c,\varepsilon}$ is shift invariant:

$$\mathcal{W}_{c,\varepsilon}(A + X, A + Y) = \int_{-\infty}^{\infty} \Pi^*(x - A, y - A) \log \frac{\Pi^*(x - A, y - A)}{\frac{1}{\mathcal{Z}} e^{-\frac{c(x-A,y-A)}{\varepsilon}}} \, \mathrm{d}x \, \mathrm{d}y \tag{41}$$
$$= \mathcal{W}_{c,\varepsilon}(X, Y).$$

According to the equation of $\overline{\mathcal{W}}_{c,\varepsilon}$, it holds the same properties as $\mathcal{W}_{c,\varepsilon}$, i.e., shift invariant and scale sensitive. Thus, we derive the convergence of distributional Bellman operator $\mathfrak{T}^\pi$ under the supreme form of $\overline{\mathcal{W}}_{c,\varepsilon}$, i.e., $\overline{\mathcal{W}}^\infty_{c,\varepsilon}$:

$$\begin{aligned}
&\overline{\mathcal{W}}^\infty_{c,\varepsilon}(\mathfrak{T}^\pi Z_1, \mathfrak{T}^\pi Z_2) \\
&= \sup_{s,a} \overline{\mathcal{W}}_{c,\varepsilon}(\mathfrak{T}^\pi Z_1(s,a), \mathfrak{T}^\pi Z_2(s,a)) \\
&= \overline{\mathcal{W}}_{c,\varepsilon}(R(s,a) + \gamma Z_1(s',a'), R(s,a) + \gamma Z_2(s',a')) \\
&\leq \overline{\mathcal{W}}_{c,\varepsilon}(Z_1(s',a'), Z_2(s',a')) + \Delta^{-k_\alpha}_{s',a',s,a}(\gamma) \\
&\leq \sup_{s',a'} \overline{\mathcal{W}}_{-k_\alpha,\varepsilon}(Z_1(s',a'), Z_2(s',a')) + \sup_{s,a,s',a'} \Delta^{-k_\alpha}_{s',a',s,a}(\gamma) \\
&= \overline{\mathcal{W}}^\infty_{-k_\alpha,\varepsilon}(Z_1, Z_2) + \Delta(\gamma)
\end{aligned} \tag{42}$$

where the first inequality comes from the scale sensitivity proof, and we denote $\sup_{s,a,s',a'} \Delta^{-k_\alpha}_{s',a',s,a}(\gamma) = \Delta(\gamma)$ for short. Since $\Delta(\gamma) \to 0$ as $\gamma \to 1$, we can conclude that $\mathfrak{T}^\pi$ is **closely** a non-expansive operator regardless of the cost function form $c$ when $\varepsilon \in (0, \infty)$. The $\gamma$-contraction of the expectation of $Z^\pi$ can be similarly proved as the KL divergence in Lemma 1. $\qquad\square$

# C  Proof of Proposition 1 and Corollary 1

*Proof.* As we leverage $\Pi^*$ to denote the optimal $\Pi$ by evaluating the Sinkhorn divergence via $\min_{\Pi \in \mathbf{\Pi}(\mu,\nu)} \overline{\mathcal{W}}_{c,\varepsilon}(\mu, \nu; k)$, the Sinkhorn divergence can be composed in the following form:

$$\begin{aligned}
&\overline{\mathcal{W}}_{c,\varepsilon}(\mu, \nu; k) \\
&= 2\mathrm{KL}\left(\Pi^*(\mu,\nu)|\mathcal{K}_{-k}(\mu,\nu)\right) - \mathrm{KL}\left(\Pi^*(\mu,\mu)|\mathcal{K}_{-k}(\mu,\mu)\right) - \mathrm{KL}\left(\Pi^*(\nu,\nu)|\mathcal{K}_{-k}(\nu,\nu)\right) \\
&= 2(\mathbb{E}_{X,Y}\left[\log \Pi^*(\mu,\nu)\right]) + \frac{1}{\varepsilon}\mathbb{E}_{X,X'}\left[c(X,Y)\right]) - (\mathbb{E}_{X,X'}\left[\log \Pi^*(\mu,\nu)\right]) + \frac{1}{\varepsilon}\mathbb{E}_{X,Y}\left[c(X,Y)\right]) \\
&\quad - (\mathbb{E}_{Y,Y'}\left[\log \Pi^*(\nu,\nu)\right]) + \frac{1}{\varepsilon}\mathbb{E}_{Y,Y'}\left[c(Y,Y')\right]) \\
&= \mathbb{E}_{X,X',Y,Y'}\left[\log \frac{(\Pi^*(X,Y))^2}{\Pi^*(X,X')\Pi^*(Y,Y')}\right] + \frac{1}{\varepsilon}(\mathbb{E}_{X,X'}\left[k(X,X')\right] + \mathbb{E}_{Y,Y'}\left[k(Y,Y')\right] - 2\mathbb{E}_{X,X'}\left[k(X,Y)\right]) \\
&= \mathbb{E}_{X,X',Y,Y'}\left[\log \frac{(\Pi^*(X,Y))^2}{\Pi^*(X,X')\Pi^*(Y,Y')}\right] + \frac{1}{\varepsilon}\mathrm{MMD}^2_{-c}(\mu,\nu)
\end{aligned} \tag{43}$$

where the cost function $c$ in the Gibbs distribution $\mathcal{K}$ is minus Gaussian kernel, i.e., $c(x,y) = -k(x,y) = e^{-(x-y)/(2\sigma^2)}$. Till now, we have shown the result in Corollary 1.

Next, we use Taylor expansion to prove the moment matching of MMD. Firstly, we have the following equation:

$$
\begin{aligned}
\mathrm{MMD}^2_{-c}(\mu, \nu) &= \mathbb{E}_{X,X'}\left[k(X,X')\right] + \mathbb{E}_{Y,Y'}\left[k(Y,Y')\right] - 2\mathbb{E}_{X,X'}\left[k(X,Y)\right] \\
&= \mathbb{E}_{X,X'}\left[\phi(X)^\top \phi(X')\right] + \mathbb{E}_{Y,Y'}\left[\phi(Y)^\top \phi(Y')\right] - 2\mathbb{E}_{X,X'}\left[\phi(X)^\top \phi(Y)\right] \quad (44)\\
&= \mathbb{E}\|\phi(X) - \phi(Y)\|^2
\end{aligned}
$$

We expand the Gaussian kernel via Taylor expansion, i.e.,

$$
\begin{aligned}
k(x,y) &= e^{-(x-y)^2/(2\sigma^2)} \\
&= e^{-\frac{x^2}{2\sigma^2}} e^{-\frac{y^2}{2\sigma^2}} e^{\frac{xy}{\sigma^2}} \\
&= e^{-\frac{x^2}{2\sigma^2}} e^{-\frac{y^2}{2\sigma^2}} \sum_{n=0}^{\infty} \frac{1}{\sqrt{n!}}(\frac{x}{\sigma})^n \frac{1}{\sqrt{n!}}(\frac{y}{\sigma})^n \quad (45)\\
&= \sum_{n=0}^{\infty} e^{-\frac{x^2}{2\sigma^2}} \frac{1}{\sqrt{n!}}(\frac{x}{\sigma})^n e^{-\frac{y^2}{2\sigma^2}} \frac{1}{\sqrt{n!}}(\frac{y}{\sigma})^n \\
&= \phi(x)^\top \phi(y)
\end{aligned}
$$

Therefore, we have

$$
\begin{aligned}
\mathrm{MMD}^2_{-c}(\mu, \nu) &= \sum_{n=0}^{\infty} \frac{1}{\sigma^{2n} n!} \left( \mathbb{E}_{x \sim \mu}\left[e^{-x^2/(2\sigma^2)} x^n\right] - \mathbb{E}_{x \sim \nu}\left[e^{-y^2/(2\sigma^2)} y^n\right] \right)^2 \\
&= \sum_{n=0}^{\infty} \frac{1}{\sigma^{2n} n!} \left( \tilde{M}_n(\mu) - \tilde{M}_n(\nu) \right)^2
\end{aligned}
\quad (46)
$$

$\tilde{M}_n(\mu) = \mathbb{E}_{x \sim \mu}\left[e^{-x^2/(2\sigma^2)} x^n\right]$, and similarly for $\tilde{M}_n(\nu)$. The conclusion is the same as the moment matching in [17]. Finally, due to the equivalence of $\overline{\mathcal{W}}_{c,\varepsilon}(\mu, \nu)$ after multiplying $\varepsilon$, we have

$$
\begin{aligned}
\overline{\mathcal{W}}_{c,\varepsilon}(\mu, \nu; k) &:= \mathrm{MMD}^2_{-c}(\mu, \nu) + \varepsilon \mathbb{E}\left[\frac{(\Pi^*(X,Y))^2}{\Pi^*(X,X')\Pi^*(Y,Y')}\right] \\
&= \sum_{n=0}^{\infty} \frac{1}{\sigma^{2n} n!} \left( \tilde{M}_n(\mu) - \tilde{M}_n(\nu) \right)^2 + \varepsilon \mathbb{E}\left[\frac{(\Pi^*(X,Y))^2}{\Pi^*(X,X')\Pi^*(Y,Y')}\right],
\end{aligned}
\quad (47)
$$

This result is also equivalent to Theorem 1, where $\Pi^*$ would degenerate to $\mu \otimes \nu$ as $\varepsilon \to +\infty$. In that case, the first regularization term would vanish, and thus the Sinkhorn divergence degrades to a MMD loss, i.e., $\mathrm{MMD}^2_{-c}(\mu, \nu)$.

$\square$

# D  Human-normalized Scores

Our implemnetation is based on [25] and all the experimental settings, including parameters are identical to the distributional RL baselines implemented by [25]. The main results about mean and

| | **Mean** | **Median** | **>Human** | **>DQN** |
|---|---|---|---|---|
| DQN | 173 % | 49 % | 17 | 0 |
| C51 | 309 % | 77 % | 26 | 42 |
| QR-DQN-1 | 430 % | 104 % | 31 | 47 |
| MMDQN | **600** % | 94 % | 27 | 43 |
| SinkhornDRL | 570 % | 89 % | 27 | 42 |

Table 2: Mean and median of best human-normalized scores across 55 Atari 2600 games. The results for all considered algorithms are aaveraged over 3 seeds.

median human-normalized scores of all considered distributional RL algorithms are reported in Table 2. Note that our implementaion is based on Pytorch, and thus the results in Table 2 are not exactly same as results implemented based on Dopamine framework [4]. However, Table 2 also suggests that our SinkhornDRL algorithm can achieve almost state-of-the-art performance in terms of mean human-normalized scores. We argue that although it seems that SinkhronDRL is on par with MMD across all games, our algorithm significant outperforms MMDDRL on a large amount of Atari games, as suggested in Figure 2. The detailed comparison based on learning curves is also exhibited in Appendix E.

# E   More experimental Results

We provide learning curves of DQN, QRDQN, C51, MMD and SinkhornDRL algorithms on all 55 Atari games in Figures 4 5 6 7 8 9. It illustrates that SinkhornDRL dramatically surpasses the other distributional RL algorithms on a large amount of environments, e.g., Venture, Atlantis, Tennis and SpaceInvader, and presents competitive performance or is only slightly inferior as opposed to the state-of-the-art baselines on other games. Note that the average improvement of SinkhornDRL on Venture game is significant owing to one to two times convergence of SinkhornDRL algorithm over 3 seeds, while the other baselines do not converge over the considered seeds. Although this improvement may also suffer from the instability issue, its occasional success for our SinkhornDRL algorithm also presents huge potential on some complicated environments. We leave the further exploration on the advantage and potential of SinkhornDRL algorithm as the future work.

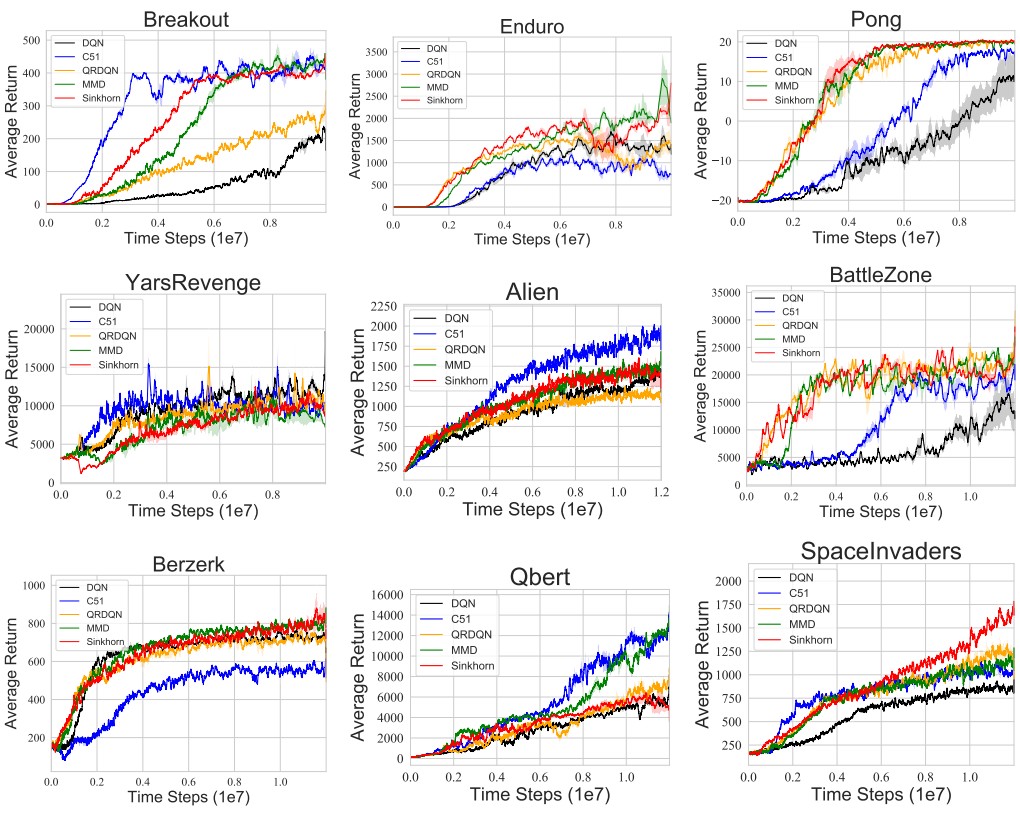

Figure 4: Performance of SinkhornDRL compared with DQN, C51, QRDQN and MMD on Breakout, Enduro, Pong, YarRevenge, Alien, BattleZone, Berzerk, Qbert and SpaceInvader.

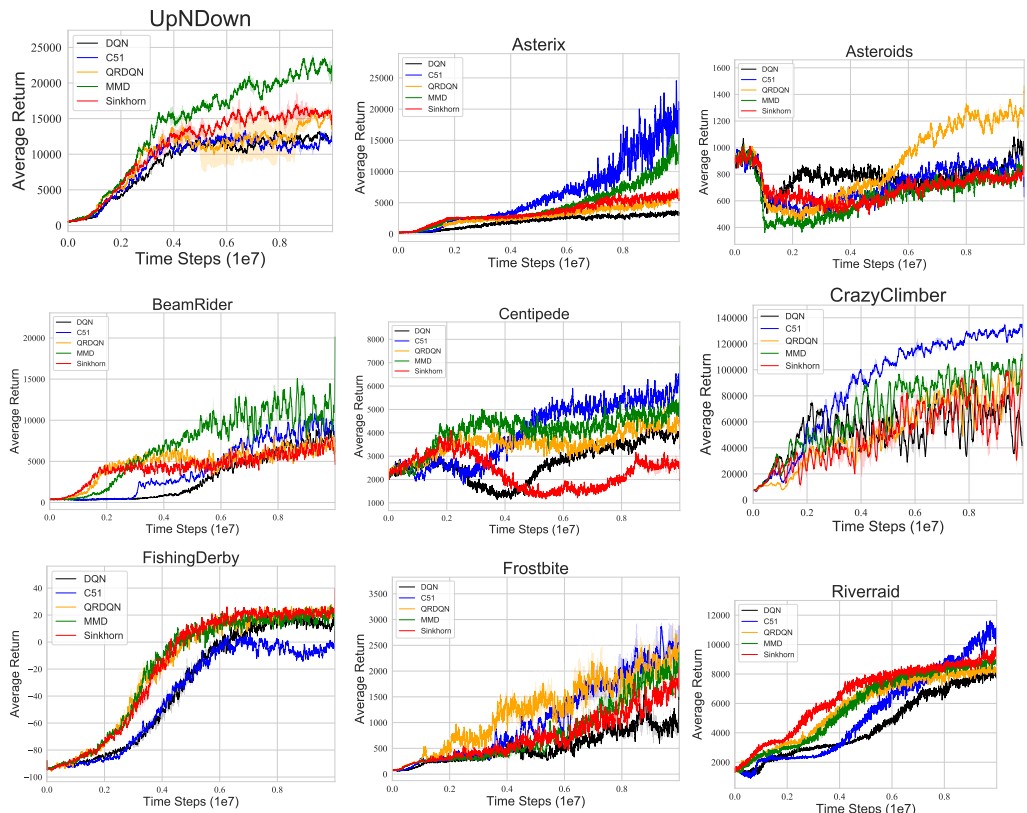

Figure 5: Performance of SinkhornDRL compared with DQN, C51, QRDQN and MMD on UpN-Down, Asterix, Asteriods, BeamRider, Centipede, FishingDerby, Frostbite and Riverraid.

## F  Sensitivity Analysis and Computational Cost

### F.1  More results in Sensitivity Analysis

From Figure 10 (a), we can observe that if we gradually decline $\varepsilon$ to 0, SinkhornDRL's performance tends to QR-DQN. Note that an overly small $\varepsilon$ will lead to a trivial almost 0 $\mathcal{K}_{i,j}$ in Sinkhorn iteration in Algorithm 2, and will cause $\frac{1}{0}$ numerical instability issue for $a_l$ and $b_l$ in Line 5 of Algorithm 2. Due to this reason, the performance of SinkhornDRL with $\varepsilon = 0.1$ or $0.075$ declines as the training proceeds, and eventually converges to the average return that QR-DQN achieves. In addition, we also conducted experiments on Seaquest, the similar result is also observed in Figure 11. The performance of SinkhornDRL is robust when $\varepsilon = 10, 100, 500$ and a small $\epsilon = 1$ tends to worsen the performance.

Moreover, for breakout, if we increase $\varepsilon$, the performance of SinkhornDRL tends to that of MMDDRL as suggested in Figure 10 (b). It is also noted that an overly large $\varepsilon$ will let the $\mathcal{K}_{i,j}$ explode to $\infty$. This also leads to numerical instability issue in Sinkhorn iteration in Algorithm 2.

In summary, the trend of SinkhornDRL to close MMDDRL and QR-DQN if we increase or decrease $\varepsilon$, respectively, provides strong empirical evidence to demonstrate the theoretical relationships between Sinkhorn divergence and MMD / Wasserstein distance, although an overly large or small $\varepsilon$ will lead to numerical instability issue.

### F.2  Comparison with the Computational Cost

We evaluate the computational time every 10,000 iterations across the whole training process of all considered distributional RL algorithms and make a comparison in Figure 12. It suggests that SinkhornDRL indeed increases around $50\%$ computation cost compared with QR-DQN and C51, but only slightly increases the the cost in contrast to MMDDRL on both Breakout and Qbert

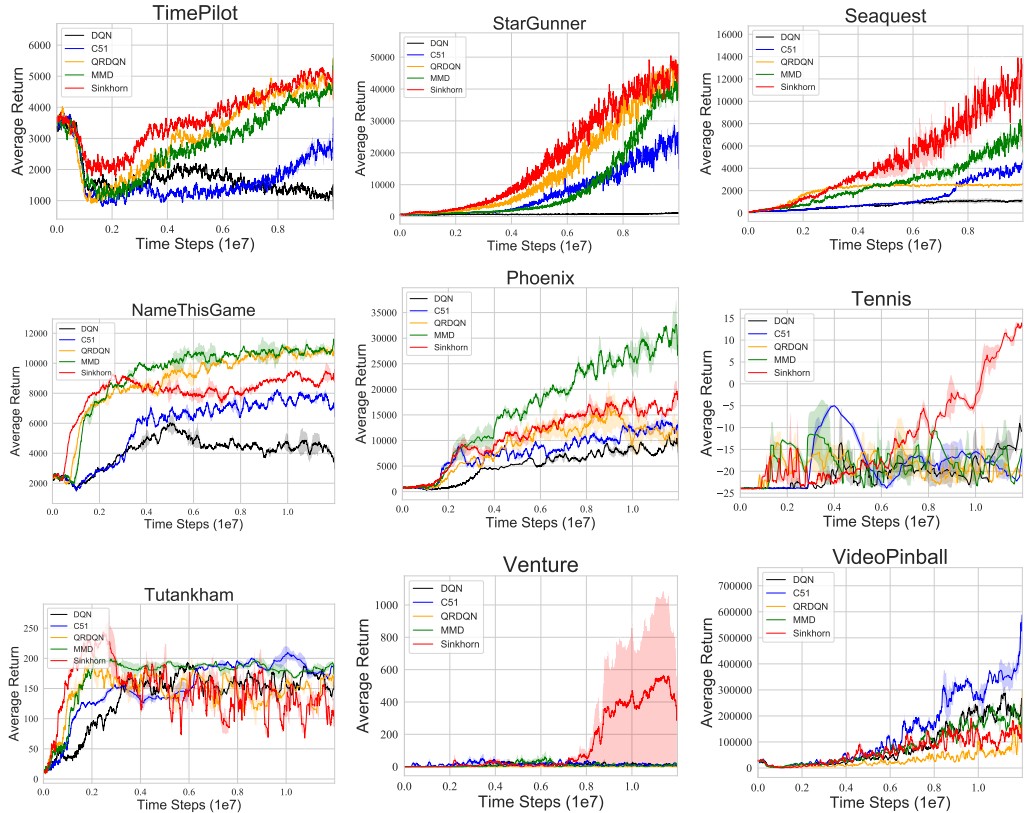

Figure 6: Performance of SinkhornDRL compared with DQN, C51, QRDQN and MMD on TimePilot, StarGuner, Seaquest, NameThisGame, Phoenix, Tennix, Tutankham, Venture and VideoPinball.

games. We argue that this additional computational burden can be tolerant in view of the significant outperformance of SinkhornDRL in a large amount of environments.

In addition, we also find that the number of Sinkhorn iterations $L$ is negligible to the computation cost, while an overly large samples $N$, e.g., 500, will lead to a large computational burden as illustrated in Figure 13. This can be intuitively explained as the computation complexity of the cost function $c_{i,j}$ is $\mathcal{O}(N^2)$ in SinkhornDRL, which is particularly heavy in computation if $N$ is large enough.

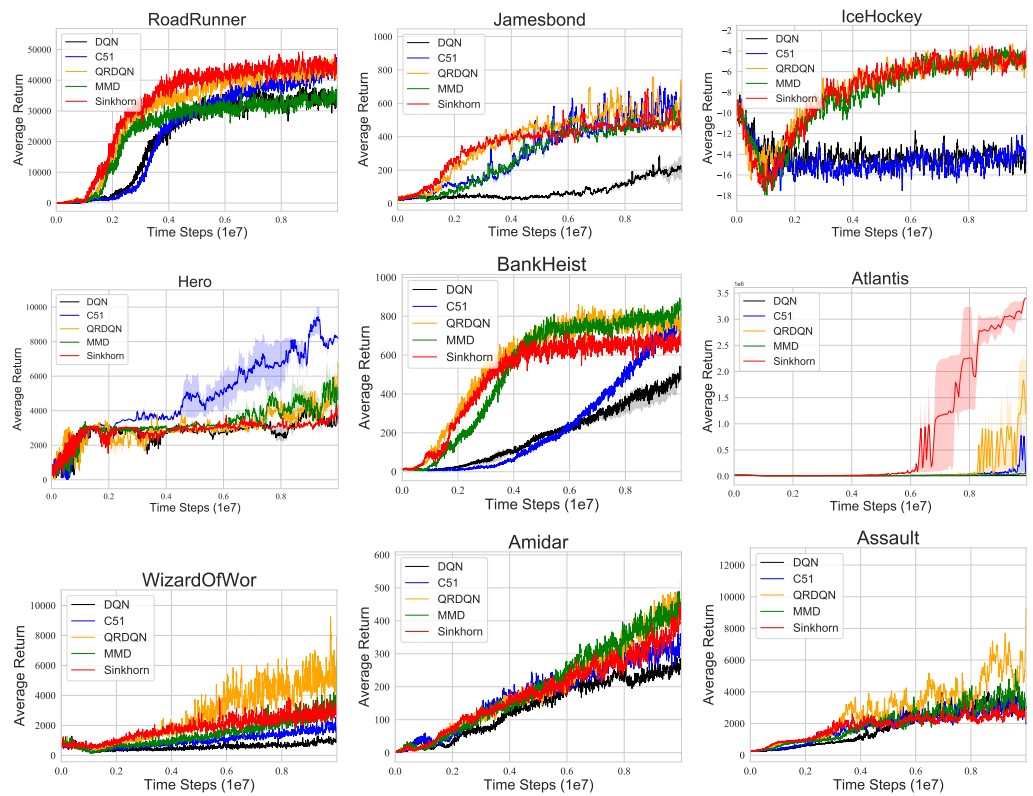

Figure 7: Performance of SinkhornDRL compared with DQN, C51, QRDQN and MMD on Road-Runner, Jamesbond, IceHockey, Hero, BankHeist, Atlantis, WizardOfWor, Amidar and Assault.

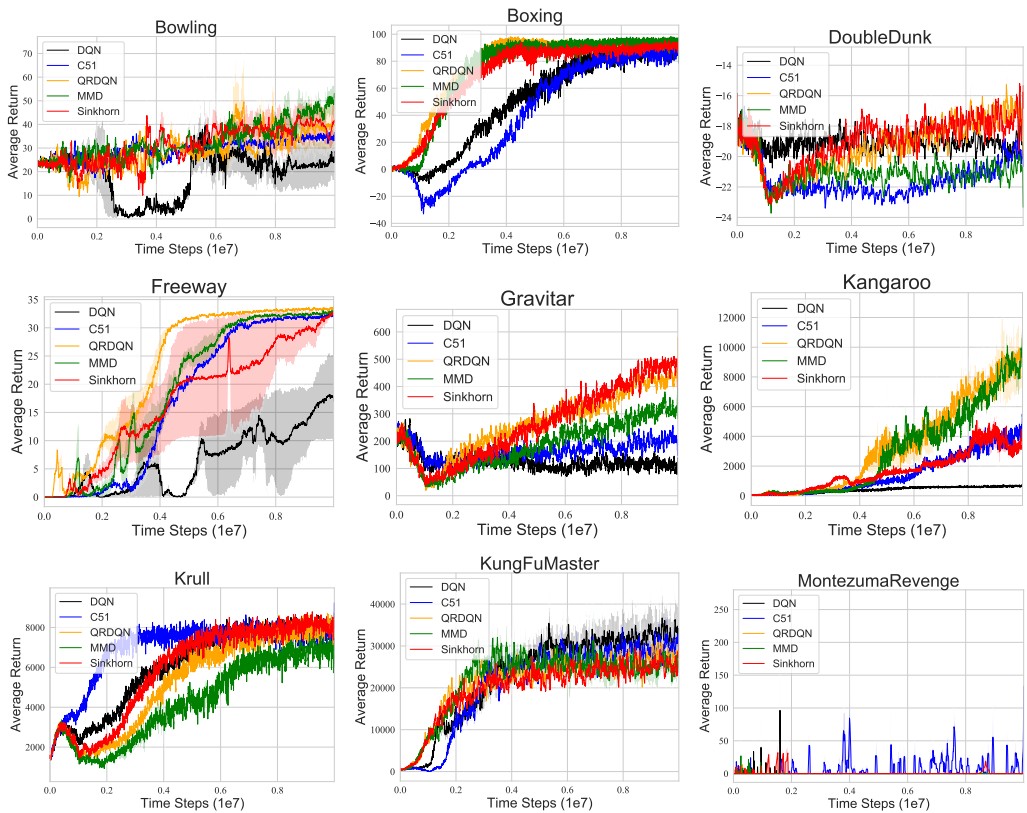

Figure 8: Performance of SinkhornDRL compared with DQN, C51, QRDQN and MMD on Bowling, Boxing, DoubleDunk, Freeway, Gravitar, Kangaroo, Krull, KunFuMaster and MontezumaRevenge.

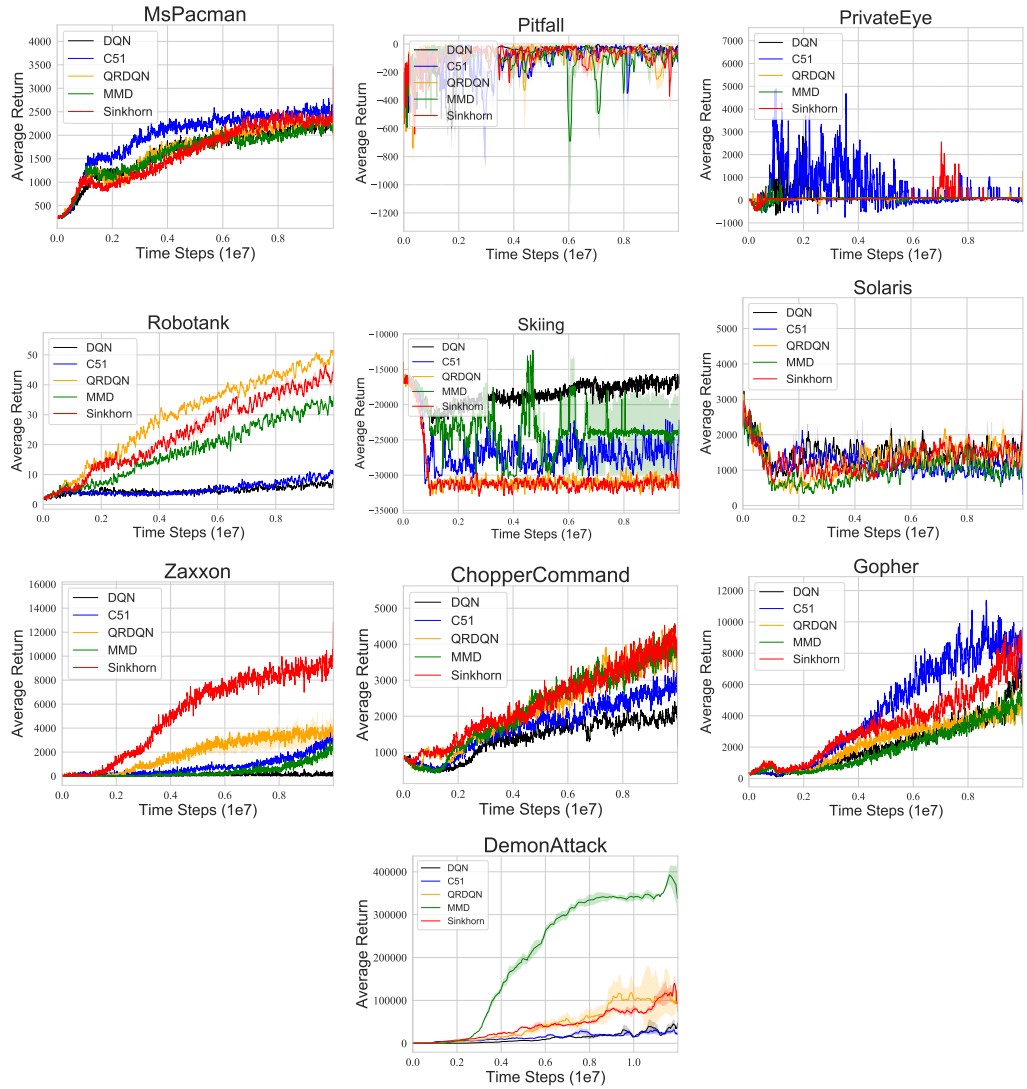

Figure 9: Performance of SinkhornDRL compared with DQN, C51, QRDQN and MMD on MsPacman, Pitfall, PrivateEye, Robotank, Skiing, Solaris, Zaxxon, ChopperCommand, Gopher and DemonAttack.

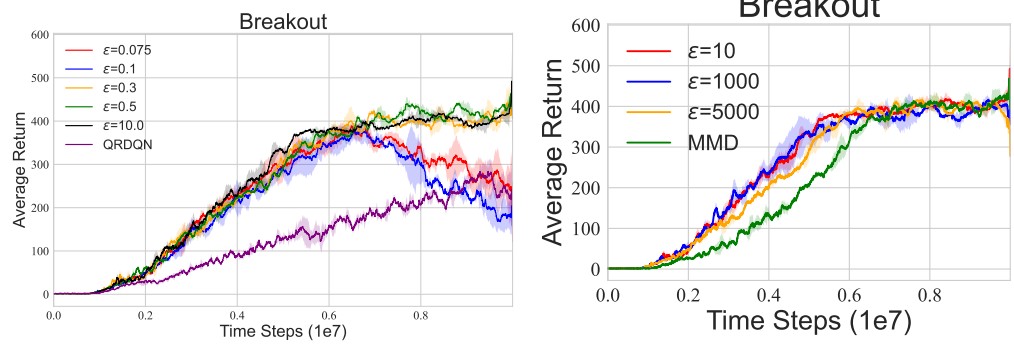

(a) Small $\varepsilon$ in SinkrhornDRL vs QRDQN     (b) Large $\varepsilon$ in SinkrhornDRL vs MMDDRL

Figure 10: (Left) Sensitivity analysis w.r.t. a small level of $\varepsilon$ SinkhornDRL to compare with QR-DQN that approximates Wasserstein distance on Breakout. (Right) Sensitivity analysis w.r.t. a large level of $\varepsilon$ SinkhornDRL algorithm to compare with MMDDRL on Breakout. All learning curves are reported over 2 seeds.

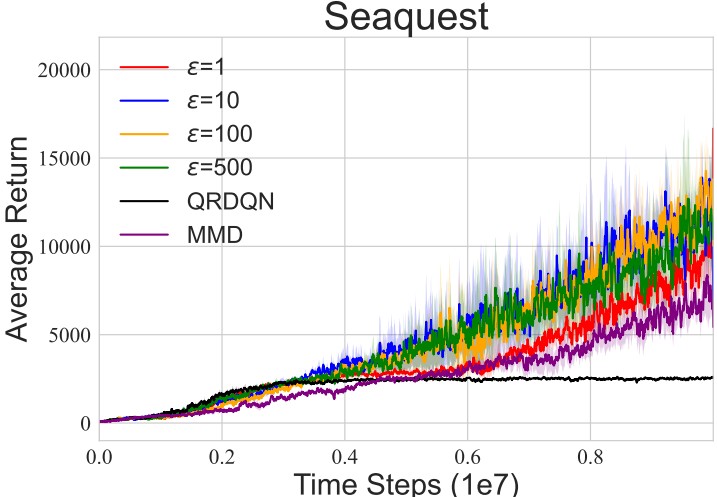

Figure 11: Sensitivity analysis w.r.t. $\varepsilon$ SinkhornDRL to compare with QR-DQN and MMD on Seaquest. All learning curves are reported over **3** seeds.

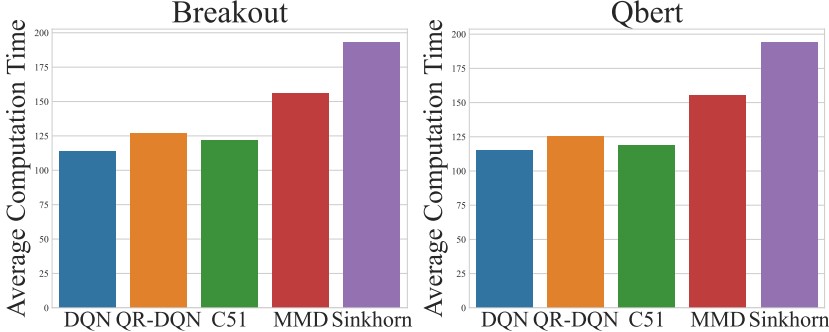

Figure 12: Average computational cost per 10,000 iterations of all considered distributional RL algorithm, where we select $\varepsilon = 10$, $L = 10$ and number of samples $N = 200$ in SinkhornDRL algorithm.

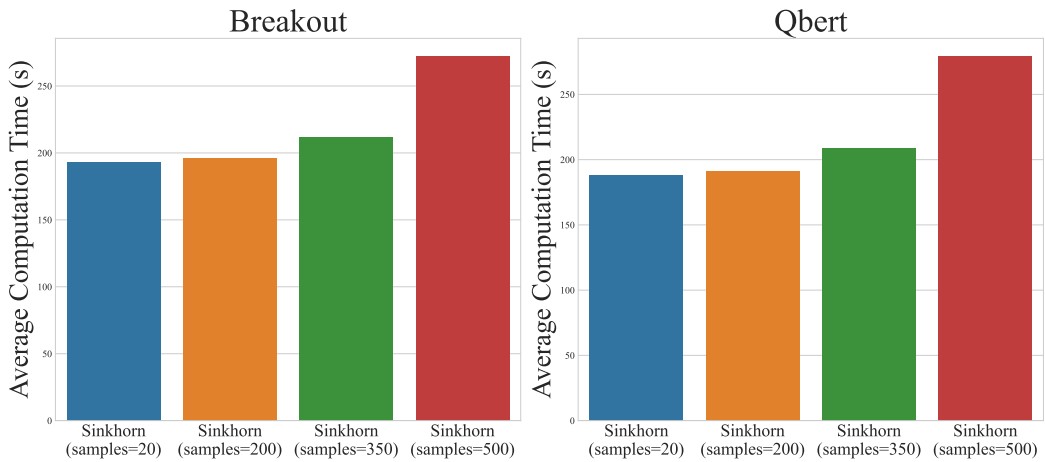

Figure 13: Average computational cost per 10,000 iterations of SinkhornDRL algorithm over different samples.