# OpenReview forum: "Distributional Reinforcement Learning via Sinkhorn Iterations"
_NeurIPS.cc/2022/Conference — NeurIPS 2022 Submitted_

### Official Review · Reviewer_pZHW · 2022-07-07

**Rating:** 6
**Confidence:** 3
**Soundness:** 3 good
**Presentation:** 3 good
**Contribution:** 2 fair

**Summary:**

The paper proposes to use the Sinkhorn divergence in distributional RL (DRL),  a relatively novel framework for RL that uses the full action-value distribution rather than its expected value, the Q-function, and that has led to good results in various application domains.  The Sinkhorn divergence has been studied previously in non-RL ML settings, and was shown to interpolate between optimal transport measures and the MMD divergence, both of which account for the geometry of the underlying spaces, and were studied in previous DRL papers. From a theoretical perspective the authors analyze the convergence properties of the distributional Bellman operators, and establish a regularized MMD equivalent form. Empirical comparisons are provided for ATARI problems.

Following the discussion and authors' response I have increased my grade from 5 to 6, supporting the acceptance of the paper.

**Questions:**

(1) Please address the convergence issues raised for algorithm 2. (2) The notion of exploration is not mentioned in the present work, and, indeed, it seems that the performance of the algorithm on problems requiring significant explorations, such as the Montezuma revenge, are inferior. It would be nice to get some insight from the authors on this issue.

**Limitations:**

The paper, while nicely combining previous ideas, does not go much beyond previous results.

**Strengths And Weaknesses:**

**Strengths** The paper is based on previous DRL methods using the optimal transport (OT) measures and MMD, by combining them into the Sinkhorn divergence, as done previously in different ML contexts (e.g., [10]). Their main theoretical contribution is item 3 in Theorem 1, demonstrating the contraction properties of the Sinkhorn divergence (items 1 and 2 follow directly from previous work, and are stated in [11]). On the empirical side, they demonstrate competitive results on the ATARI tasks.

**Weakness**
The main issue with the paper, as far as I see it, pertains to its level of novelty w.r.t. previous work. It essentially combines known previous ideas (the Sinkhorn divergence as an interpolant between OT and MMD measures, and DRL), and suggests a new DRL algorithm that takes advantage of the combined merits. Importantly, the main merit of using the Sinkhorn measure, its lack of dependence on the dimension, appears already in [11].
If I understood correctly, there is no proof that Algorithm 2 converges, or bounding its finite sample error for a given value of L. This is an important issue in the overall convergence of the algorithm. Below Algorithm 2 the authors state that “It suggests that the Sinkhorn iteration can asymptotically converge to the true loss in a linear rate” – what does ‘suggest’ mean? What does ‘can converge’ imply? Under what conditions?

The authors refer several times to the ‘unbiased gradient estimate of MMD’ that leads to superior performance. However, this is never discussed or explained, while playing a major role in their arguments.

Re the empirical results, it would be good if the authors discussed the exploratory properties of the algorithm, vis-à-vis other DRL methods, as these play a key role in many challenging RL tasks. For example, it appears that in the Montezuma Revenge game, that is known to require careful exploration, they do worse than other DRL approaches.

**Minor points** Section 2.2 does not distinguish between scalar and vector random variables (as is done in the appendix). For example, eq. (3) holds only for scalars. Even if they only use this case, this point should be mentioned. In eq. (4) the MMD should include arguments (distributions or RVs). Also, X’ and Y’ are not defined.

Algorithm 1 is described as an “RL Algorithm”. However, it is not a full RL algorithm, but, rather, just the update step. For the reader’s sake, it would be clearer if the authors presented pseudo-code for the full RL algorithm, even if parts of it are standard.

---

> ### Author Response · Authors · 2022-08-01
> **Response**
>
> Thank you for appreciating our work. Below we address all your concerns. Please kindly let us know if there are any further questions.
>
> ### Question 1: novelty issue
> **research issue novelty**. As a norm in distributional RL, Wasserstein distance, albeit being tricky to directly evaluate, exhibits huge empirical success e.g., QR-DQN. Sinkhorn divergence, which is renowned as a successful norm to approximate Wasserstein distance, has not been studied in distributional RL setting. Thus, it is definitely novel to explore the efficacy of ``intermediate'' Sinkhorn divergence in the distributional RL setting.
>
> **theory novelty**. Theorem (1) and (2) can be viewed as the follow-up conclusion in terms of the convergence (rate) of distributional Bellman operators based on the theoretical relationships between MMD, Sinkhorn and Wasserstein distance. More importantly, our main contribution is (3), where we view Sinkhorn divergence as an optimal KL divergence based and thus the non-expansion property is derived. Besides, the regularized MMD relationship is also another theory novelty beyond the existing conclusion.
>
> **algorithm novelty**. We think SinkhornDRL is a nice central algorithmic contribution. The Sinkhorn approach to distributional RL is novel and is likely to be of interest to the deep RL community
>
> ### Question 2: convergence of Sinkhorn iteration
> We would like to point out that the convergence of Sinkhorn iterations is a known fact in the advanced linear algebra literature[1], which is also further discussed in [2]. In particular, using Hilbert’s projective metric, Franklin and Lorenz[1] proved that the convergence of the scaling factor in Sinkhorn iteration is linear. Please refer to [1,2] for more rigorous proof. We also revised our claim in the updated version based on your suggestion.
>
> ### Question 3: unbiased gradient estimate of MMD
> As pointed out in [1], Wasserstein distance does not enjoy unbiased gradient estimate property, which may hurt the performance in certain games. By contract, MMD and the limiting case of SinkhornDRL enjoy this property, and thus the superiority of MMDDL or SinkhornDRL can be partially attributable to this desirable property. As it is a known conclusion based on the optimal transport or divergence literature, we have not provided too many details on this argument.
>
> ### Question 4: exploratory behaviors of SinkhornDRL
> We would like to clarify that in Montezuma all the typical DRL algorithms perform poorly not merely for SinkhornDRL as shown in Figure 8. Note that the superiority of distributional RL algorithms, including QR-DQN, MMD, heavily depends on the characteristics of environments, which is still an open problem to explain whether a specific algorithm outperforms others in certain games or not.  In our opinion, SinkhornDRL can both enjoy the geometry property of Wasserstein distance and unbiased gradient estimate for MMD, and this nice trade-off can potentially yield remarkable superiority in certain games, where the value distribution might be more complicated to represent and reducing gradient bias can be substantially helpful to the optimization. We leave more rigorous investigation towards this issue in the future.
>
> ### Question 5: Minors
> We revised typos, definition and caption of Algorithm in the updated version based on your suggestions.
>
> [1] Franklin, J. and Lorenz, J. (1989). On the scaling of multidimensional matrices. Linear Algebra and its applications, 114:717–735.
>
> [2] Cuturi, Marco. "Sinkhorn distances: Lightspeed computation of optimal transport." Advances in neural information processing systems 26 (2013).

---

> > ### Comment · Reviewer_pZHW · 2022-08-08
> > **Questions**
> >
> > Thank you for your response to my review and to others, based on which I believe that the paper is less incremental than I had originally perceived it to be. Re the unbiased gradient estimate of MMD, since this is an essential feature it is important to provide a precise reference to this issue, even it is known. I will consider raising my grade, once I am convinced that the correctness issues of the proof of Theorem 1, as raised mainly by reviewer ptV9, are fully settled.

---

### Official Review · Reviewer_ptV9 · 2022-07-07

**Rating:** 5
**Confidence:** 5
**Soundness:** 2 fair
**Presentation:** 2 fair
**Contribution:** 3 good

**Summary:**

This paper presents a new approach to distributional reinforcement learning based on regularized optimal transport. The authors present an implementable algorithm based on Sinkhorn iterations for approximating the Sinkhorn divergence, and investigate the performance of this approach on the Atari suite of games.

---

I thank the authors for their engagement during the discussion. The updates made during this period have improved the paper in my view, particularly regarding the main theoretical result. I still have two key reservations about this result in its current form, which was uploaded shortly before the end of the discussion period:
 - First, the statement is not presented clearly: "T^pi is a closely non-expansive operator" is not a precise mathematical statement. Reading the appendix proof means it can be inferred what is meant here, but in terms of clarity I found this to be poor.
 - Second, the proof is not written clearly, and this makes checking the proof more difficult as well, and I am not 100% confident about the validity of the proof. Aside from a general lack of clarity in the mathematical argument, some of the issues include:
    - Assuming the existence of densities.
    - Some typos (Eqns (39), (40), 1/Z_1 should be 1/Z_2; equation below Line 518 is missing an implication sign; Line 510 K(2U,2V) should be K(aU, aV)).
    - In the proof, it is apparent that the term Delta^c_{mu, nu} depends on the two distributions mu and nu, meaning that the operator is not uniformly "almost a contraction", but this is not made clear in the theorem statement, which is potentially misleading.

My current standing is that the core algorithmic contribution is interesting, although I still have reservations about the quality of the presentation (including the description above) and the experimental results.

**Questions:**

Theorem 1: (1) & (2). These results are making limiting statements as eps -> 0, infty, but then seem to be drawing conclusions about general epsilon, which I don't think is valid; perhaps there is a typo here? If these are just statements about the limited values eps=0, and eps->infty, it should be made clearer that these results are not original to this paper. This is the same for the expectation part of (3), which I think is attributable to Bellemare et al. (2017).

Part (3) seems to contradict the counterexample given by Nguyen et al. (2020). More specifically, Nguyen et al. provide an example in which sup-MMD_k(T^pi Z_1, T^pi Z_2) > sup-MMD_k(Z_1, Z_2). Given that the Sinkhorn divergence approaches the MMD as eps -> infty, this example must also apply to the case of Sinkhorn divergences for sufficiently large epsilon, so I don't think it is true that the distributional Bellman operator is a non-expansion under Sinkhorn divergences in general.

Looking at the proof, I think there are a couple of steps which aren't valid/require more care:
 - Equation (28). It seems to be forgotten here that s', a' are random variables, and so the inequality that appears in this sequence of equations requires further justification.
 - Equation (30). I think one of the key issues here is that Pi^* depends on the distributions of the input random variables in a potentially complicated way, while I think the claimed derivation here assumes that Pi* for aX and aY is obtainable through a simple change of variables from that of X and Y.

In summary, I think part (3) of Theorem 1 is not correct as stated.

Other questions/notes:

It is unclear to me what Table 1 is intended to communicate. The convergence rates/contraction rates stated are with respect to different metrics, so these are difficult to compare. I was also unsure the relevance of the sample complexity column it looks as though these are upper-bounds on expected error in estimating the metric with empirical distributions, but it is unclear to me why this is relevant to distributional reinforcement learning, where generally single transition samples are used. These bounds are also stated with general d in some cases, whereas d=1 in distributional reinforcement learning. Can the authors clarify what readers are intended to take from this table?

Line 289: The authors state "SinkhronDRL only slightly increases the computational cost", but Figure 10 in Appendix F seems to indicate that SinkhornDRL is around 50% slower than baselines such as QR-DQN, which is arguably more than a slight increase. It would have been interesting to see how dependent this is on the value of L too. The sensitivity analysis indicates (on Breakout) that the approach is robust to taking L as low as 2. Does this bring the computational cost in line with other distributional RL approaches? Does this robustness generalize to other games?

In general I found the experiments to be clearly presented, although further information on hyperparameter sweeps, preliminary runs, and selection would help improve their interpretability; can the authors provide this information? Were 10M frames (rather than 200M) used for computational/resource reasons (this is not necessarily a problem if so)? How were the 9 games in Figure 1 selected for the main paper?

There are some typos to address throughout the paper. Some examples are:
Line 8: "Bellmen" -> "Bellman"
Line 19: Duplicate reference 7 -> I expect one of these is meant to be [6]?
Line 205: "Proof of Corollary" -> "Proof of Corollary 1"
Line 219: "differential" -> "differentiable"?

Minor technical notes:
Equation (1): Should be equal in distribution.
Line 174: "General case eps in (0, eps)". This is presumably a typo? Can the authors clarify what is meant here?

**Limitations:**

The inclusion of the computational cost comparison in Appendix F is very useful, although I think the comparison is not currently accurately reflected in the main text (see above).

**Strengths And Weaknesses:**

Overall, I think there is a nice central algorithmic contribution in this paper. The Sinkhorn approach to distributional RL is novel as far as I am aware, and is likely to be of interest to the deep RL community at NeurIPS. I think there are some instances of incorrect results in the paper currently, which means I don't think the paper can be accepted currently, but I would be happy to revise my rating if the authors can address this, or potentially remove the incorrect parts of the results if they can't be resolved.

Strengths:
 - In my view, this is an interesting, unexplored approach to distributional reinforcement learning, which interpolates between two known approaches, based on Wasserstein distances and MMD.
 - The implemented algorithm performs well in Atari, generally comparing favorably to several existing approaches to distributional RL, and this approach is likely to be of interest to deep reinforcement learning researchers.
 - In addition to the core Atari results, the authors also include a sensitivity analysis regarding the two key hyperparameters their method uses, demonstrating that the method is fairly robust to the choice of these hyperparameters.

Weaknesses:
 - I think Theorem 1 as stated is not true. More details are given below. This is partly the reason for a "reject" recommendation currently, but I would be very happy to revise this rating if the authors can address the concerns.
 - The experimental evaluation setting is slightly non-standard, training for 10M rather than 200M frames on Atari. I couldn't find information about how hyperparameters were selected, e.g. were any sweeps/preliminary runs carried out? This is important information for properly interpreting the reported results.
 - I think there are currently some overly-strong claims made in the paper (e.g. "SinkhronDRL only slightly increases the computational cost", see below) that the authors could consider revising.

---

> ### Author Response · Authors · 2022-08-01
> **Response**
>
> Thank you for your valuable suggestions. Below we address all your concerns. Please kindly let us know if there are any further questions.
>
> ### Q1: Theorem 1 (1-3)
> The conclusion drawn from Theorem 1 (1,2) can be viewed as follow-up results based on relationships between MMD, Sinkhorn divergence and Wasserstein distance, as we further provide a compressive proof regarding the impact of limiting behavior of Sinkhron divergence on the convergence (rate) of distributional Bellman operator. We think the main contribution is the part (3) when $\varepsilon \in (0, \infty)$, where we cast the Sinkhorn divergence as a specific KL divergence between an optimal joint distribution and a Gibbs distribution determined by its margins. Although a similar result of expectation part (3) w.r.t. non-expansion property can be found in Bellemare et.al, we firstly offer rigorous proof from the perspectives of shift-invariant(I), scale sensitive(S) and unbiased gradient estimate(U) and particularly prove under the Sinkhorn divergence. We have made it clearer in the revised paper to posit the real contribution of our work.
>
> ### Q2: counterexample in MMDDRL
> We would like to clarify that the counterexample in MMDDRL is to demonstrate that distributional Bellman operator under MMD with **Gaussian kernel** is not contractive. For the general $\varepsilon$ in Sinkhorn divergence, our proof is based on the conclusion in the statistical physics (Line 137-139) that sinkhorn divergence can be re-factored as a projection problem with a KL divergence between an optimal joint distribution and a Gibbs distribution. Since distributional Bellman operator under KL divergence has been proven to be a non-expansion operator (simply mentioned in [1] without detailed proof), and thus the conclusion should be intuitive. For the case when $\varepsilon$ is sufficiently large, we think it is also reasonable and fair to say that the limiting behavior of Sinkhorn divergence could be different from $\varepsilon\in(0, \infty)$ under certain cases. MMD with Gaussian kernel is not contractive (we argue it is not quite contractive as the empirical results are desirable), which can be different from the conclusion drawn in the general case for Sinkhorn divergence (in (3) we find it is non-expansion rather than the contraction, which already serves as a cost).
>
> ### Q3: about careful proof
> For eq 28, we revised the formula in the updated version based on your suggestion, but the conclusion still holds based on the fact
> $$D_{\mathrm{KL}}\left(Z_{1}\left(S^{\prime},A^{\prime}\right),Z_{2}\left(S^{\prime}, A^{\prime}\right)\right) \leq \sup_{\mathrm{s}^{\prime}, \mathrm{a}^{\prime}} D_{\mathrm{KL}}\left(Z_{1}\left(s^{\prime}, a^{\prime}\right), Z_{2}\left(s^{\prime}, a^{\prime}\right)\right) $$
>
> For Eq.30, we know the fact that $p(aX<z_1, aY<z_2)=F(x/a, y/a)$ as long as the pdf $\Pi^*$ exists and $x, y \in(-\infty, \infty)$, regardless of the potentially complicated form of $\Pi^*$.
>
> ### Q4:Table 1
> Table 1 is intended to provide a rigorous and comprehensive comparison of all typical distributional RL algorithms as well as their key proofs in Appendix A. It conveys the fact that the choice of representation and the representation manner of $Z_\theta$ are two crucial factors as the title of Section 3.2 suggests, and corresponds to different convergence rate and sample complexity. Sample complexity i.e., approximating the distance with samples of measures, is also a typical concept in the optimal transport literature and hence we list it here in a comprehensive way. As we leverage value function NN to generate samples of each state, we still need to consider the approximation error between the true distance and the empirical version via samples, which requires us to compare the sample complexity for different divergences.
>
> ### Q5: computational cost  and typos
> We added more ablation studies in Fig 13 of Appendix F in the revised paper. It suggests that decreasing $L$ is negligible to saving computational cost and an overly large $N$ can lead to large overhead.
>
> It is true that SinkhornDRL indeed increases by around 50% overhead compared with QRDQN and C51, but the cost slightly increases compared with MMD (by around 20%). We have revised our claims in the revised paper.
>
> We choose 100M frames for the reason of saving computational cost. For the hyper-parameters sweep, we did it on Breakout and find the performance is not sensitive to L in a proper range and $\varepsilon$. We finally choose $L=10$, $N$=200, and $\varepsilon=200$ across all games. It is well-expected that a more careful choice of these hyper-parameters for each game can yield better results. A similar result on Seaquest is given in Appendix F.
>
> **We corrected the typos and issues in the revised paper and provide clarifications for your concerns. It is much appreciated if you could raise your rating**.
>
> [1] Dabney, Will, et al. "Distributional reinforcement learning with quantile regression." AAAI 2018

---

> > ### Comment · Reviewer_ptV9 · 2022-08-05
> > **Thank you for your response: follow-up questions**
> >
> > I thank the authors for their response. I have a few follow-up queries on particular parts of it.
> >
> > Q1.
> > "Although a similar result of expectation part (3) w.r.t. non-expansion property can be found in Bellemare et.al,"
> > If I understand correctly it is not just similar, but identical to their Lemma 4. Can the authors comment on this?
> >
> > In my view, parts (1) and (2) of Theorem 1 are still unclear. For example, in part (1), since the initial part of the statement relates bar{W}_{c,eps} to \mathcal{W}_\alpha (which I think is a typo and should be W_\alpha?) specifically as epsilon -> 0, the final part of the statement about a gamma-contraction should not apply to bar{W}_{c,eps} with an unspecified epsilon.
> >
> > In addition, the proof of part (1) seems to require a choice of c = - k_\alpha, so the statement of the theorem needs to be updated to make clear that the result only holds for this choice of c. Can the authors confirm that this is the case?
> >
> > Q2.
> >
> > The core problem I see with the proof at Equation (30) currently is that Pi* (which the authors mention can be viewed as the optimal KL projection of a certain joint distribution) depends on the two input distributions to \mathcal{W}_{c,eps}. That is, Pi* for the pair (aX, aY) (call it Pi_1) is clearly different to Pi* for the pair (X, Y) (call it Pi_2), and moreover there is not necessarily a simple relationship between them. However, in Equation (30), the argument seems to rely on a claim such as Pi_1(x,y) \propto Pi_2(x/a,y/a), which I don't think is true in general.
> >
> > A related observation is that Equation (30) claims to establish that \mathcal{W}_{c, eps}(aX, aY) = \mathcal{W}_{c, eps}(X, Y). If we take eps -> 0 with c corresponding to the rectified kernel, we would arrive at W_alpha(aX, aY) = W_alpha(X, Y), which is not true, and likewise if we take c corresponding to the Gaussian kernel and let eps -> infty, we would obtain MMD_Gaussian(aX, aY) = MMD_Gaussian(X, Y), which is also not true.
> >
> > I think this will present problems if we take e.g. c to correspond to the Gaussian kernel, because for high enough epsilon, then we have \mathcal{W}_{c, eps} -> MMD_Gaussian as eps -> infty, so the same counterexample to non-expansion that works for the Gaussian kernel will also work for \mathcal{W}_{c, eps} with high enough epsilon.
> >
> > Please let me know if I am mistaken in any of the above, and whether the authors agree that there may be an issue in the proof at Equation (30).

---

> > > ### Author Response · Authors · 2022-08-06
> > > **Thank you for follow-up questions. Response to Q1 (1/2)**
> > >
> > > Thanks for the follow-up questions and we hope our following response can further clarify your concerns. We are also happy to respond to any further questions.
> > >
> > > ### Response to Q1
> > >
> > > **Expectation part (3).** We mainly contribute to making the conclusion of the non-expansion operator in (3) (more proof details are provided in Q2). Immediately, we added a quick follow-up statement regarding the expectation behavior, which, as you said, indeed can be directly based on Lemma 4 in Bellemare's paper. However, we view it more as quick evidence to explain its empirical success rather than our main theoretical contribution. We revised our claim and added the citation in the updated version.
> > >
> > > **Parts (1) and (3) for unspecified $\varepsilon$.** Thank you for pointing out this issue. Our conclusion is initially for the limiting behavior of $W_{c, \varepsilon}$ when $\varepsilon = 0$, but you are right this conclusion may not be rigorous enough for the unspecified $\varepsilon$ when we let $\varepsilon \rightarrow 0$ rather than exactly $=0$. However, we provide the following rigorous proof to show that, for example in part (1), this distance difference is **at most an infinitesimal quantity $\mathcal{O}(\varepsilon)$** and it will vanish when $\varepsilon \rightarrow 0$.  Firstly, as $W_{c, \varepsilon} \rightarrow W_\alpha$ (as you pointed out, there is a typo) and the regularization term is non-negative, using the language of $(\varepsilon, \delta)$ definition,  we have: for any $\varepsilon$, there exists a $\delta(\varepsilon)$, such that $W_{c, \varepsilon} - W_\alpha < \delta(\varepsilon)$. Since $\delta(\varepsilon) \rightarrow 0$ as $\varepsilon \rightarrow 0$, we denote $\delta(\varepsilon)$ as $\mathcal{O}(\varepsilon)$. Therefore, we have the contraction conclusion regarding the supreme form of Sinkhorn divergence $W_{-\kappa_\alpha, \varepsilon}^\infty$:
> > >
> > > $$
> > > \begin{align}
> > > W_{-\kappa_\alpha, \varepsilon}^\infty(\mathfrak{T}^\pi Z_1, \mathfrak{T}^\pi Z_2)
> > > &=W_{-\kappa_\alpha, 0}^\infty(\mathfrak{T}^\pi Z_1, \mathfrak{T}^\pi Z_2) - W_\alpha^\infty(\mathfrak{T}^\pi Z_1, \mathfrak{T}^\pi Z_2) + W_\alpha^\infty(\mathfrak{T}^\pi Z_1, \mathfrak{T}^\pi Z_2) \\\\
> > > 	& =  \mathcal{O(\varepsilon)} + W_\alpha^\infty(\mathfrak{T}^\pi Z_1, \mathfrak{T}^\pi Z_2)
> > > \end{align}
> > > $$
> > >
> > > where the second term $ W_\alpha^\infty(\mathfrak{T}^\pi Z_1, \mathfrak{T}^\pi Z_2)$ is contractive, and thus for the unspecified $\varepsilon$, the only difference from the limiting $\varepsilon=0$ is a term $\mathcal{O(\varepsilon)}$, which will vanish as $\varepsilon \rightarrow 0$. Similar proof to (3) can be easily derived as $\varepsilon \rightarrow \infty$, where we can use $\mathcal{O}(\frac{1}{\varepsilon})$ as the infinitesimal quantity.
> > >
> > > **Revised statement.** We agree with you as the definition of Wasserstein distance requires $c=-k_\alpha$ and the general $c$ refers to the general optimal transport. We updated this statement in the revised paper based on your suggestion.

---

> > > > ### Comment · Reviewer_ptV9 · 2022-08-08
> > > > **Thanks for your response**
> > > >
> > > > Thank you for your response.
> > > >
> > > > Expectation part. I can't see where an update has been made regarding the citation for the fact that the expectation-contraction result is already known. Can the authors indicate where in the paper this is?
> > > >
> > > > Unspecified epsilon. I agree with the general idea of this argument, although there are a few inaccurate details in the authors' description. First, it's important to note that (as far as I know), delta(epsilon) will also depend on the two input distributions, so we can't make a uniform choice of delta(epsilon). Second, it's not correct to write delta(eps) = O(eps) [which would imply delta(eps) converges to 0 at least as quickly as eps], so the final statement in the displayed equations ends up being slightly misleading because of this. What would be clearer would be to keep delta(eps) rather than passing to O(eps).
> > > >
> > > > There is still an issue with the way that these results are stated in Theorem 1, which is currently:
> > > >
> > > > "As eps->0 ... T^pi is a gamma-contraction under \bar{W}^\infty_{c,eps}"
> > > >
> > > > The issue is that, as the authors say above, there is no epsilon > 0 for which it is true that "T^pi is a gamma-contraction under \bar{W}^\infty_{c,eps}".

---

> > > > > ### Author Response · Authors · 2022-08-09
> > > > > **Author Response to Follow-up Question**
> > > > >
> > > > >
> > > > > We sincerely appreciate your further input and efforts to improve our paper. We revised our paper based on your suggestions.
> > > > >
> > > > > **An updated version about the expectation part.** To make it clearer and posit our original contribution, we remove the expectation part in Theorem 3 in the main text and only discuss it in the appendix because it is a corollary directly based on the previous paper. Please see the main paper or the supplementary file that includes the appendix.
> > > > >
> > > > > **Unspecified $\varepsilon$**. We totally agree with your suggestions towards a more accurate statement and thanks for pointing out this issue. We revised our paper based on your suggestion. Particularly, we keep the $\delta$ rather than passing to $\mathcal{O}(\varepsilon)$. Also, we agree $\delta$ should depend on two input variables, but it may not hurt the conclusion as all conclusions made in Theorem 1 are based on two given (arbitrary) input random variables.
> > > > >
> > > > > **Statement in Theorem 1.** We agree and thus we revised our statement in Theorem 1 and make the contraction conclusion only when $\varepsilon = 0, \infty$. For the unspecified $\varepsilon \rightarrow 0, \infty$, we put our accurate statement using the $(\epsilon, \delta)$ language in the appendix based on your suggestions.

---

> > > ### Author Response · Authors · 2022-08-06
> > > **Thank you for follow-up questions. Response to Q2 (2/2)**
> > >
> > > Thanks for the follow-up questions and we hope our following response can further clarify your concerns. We are also happy to respond to any further questions.
> > >
> > > ### Response to Q2
> > >
> > > **Different optimal joint distribution $\Pi$.** We thank you for pointing out our inaccurate proof here. After we slightly modify this proof, we obtain an inequality relationship, i.e., $W_{c, \varepsilon}(aX, aY) \leq W_{c, \varepsilon}(X, Y)$, and the final non-expansive conclusion in Theorem 1 (3) still satisfies. In particular, we agree that for $(aX, aY)$, the optimal joint distribution can be different and we did not fully realize that before. However, due to the **infimum nature** of sinkhorn divergence, $W_{c, \varepsilon}(aX, aY)$ can be less than or equal to the quantity with a **specific** joint distribution. We denote $\Pi^1$ as the optimal joint distribution for $(aX, aY)$, and $\Pi^2$ as the optimal joint distribution for $(X, Y)$. We let $X^\prime = a X$ and $Y^\prime = a Y$. Then we have:
> > >
> > > $$
> > > \begin{align}
> > > 	W_{c, \varepsilon}(aX, aY) & = \text{KL}(\Pi^1(x^\prime, y^\prime), \mathcal{K}(x^\prime, y^\prime)) \\\\
> > >         & \leq \text{KL}(\Pi^2(x^\prime, y^\prime) || \mathcal{K}(x^\prime, y^\prime)) \\\\
> > > 	&=\int_{-\infty}^{\infty} a^2 \Pi^2(ax, a y)  \log \frac{\Pi^2(ax,ay)}{\mathcal{K}(ax, ay)}  dx  dy \\\\
> > > 	&= \int_{-\infty}^{\infty} \Pi^2(ax, ay) \log \frac{ \Pi^2(ax, ay)}{\mathcal{K}(ax, ay)} d ax d ay \\\\\
> > > 	& =W_{c, \varepsilon}(X, Y)
> > > \end{align}
> > > $$
> > >
> > > Therefore, the inequality $W_{c, \varepsilon}(aX, aY) \leq W_{c, \varepsilon}(X, Y)$ can still guarantee the distributional Bellman operator is **at least** non-expansive contraction, i.e.,  $W_{c, \varepsilon}^\infty(\mathfrak{T}^\pi Z_1, \mathfrak{T}^\pi Z_2) \leq W_{c, \varepsilon}^\infty(Z_1, Z_2) $. We revised the proof in the updated version.
> > >
> > > **Consistency with Wasserstein distance.** This inequality becomes consistent with the Wasserstein distance scenario when $\varepsilon \rightarrow0$ as $W_\alpha(aX, aY)\leq |a| W_\alpha (X, Y) \leq W_\alpha(X, Y)$ when $a\leq1$ ($\gamma \leq 1$ in the common RL case).
> > >
> > > **Consistency with MMD by checking the counterexample.** For the limiting MMD case when $\varepsilon \rightarrow \infty$, after carefully checking the counterexample in MMD paper, we find the key equality in the proof to guarantee the expansion depends on the **linearity property of kernel in MMD**, but it usually does not hold for other divergences, including KL divergence, Wasserstein distance and Sinkhorn divergence. Concretely, we list the sketch of proof (Eq 40 in MMD paper) here:
> > >
> > > $$
> > > \begin{align}
> > > \operatorname{MMD}^{2}\left(\mathcal{T}^{\pi} \mu\left(s_{0}\right), \mathcal{T}^{\pi} \nu \left(s_{0}\right) ; k\right) & =\operatorname{MMD}^{2}\left(E \left[ \eta_r \right], E\left[ \kappa_r \right],  k\right) \\\\
> > > &=\sum_{r, r^{\prime} \in \operatorname{dom}(R)} p_{r} p_{r^{\prime}} \operatorname{MMD}^{2}\left(\eta_{r}, \kappa_{r^{\prime}} ; k\right) \\\\
> > > &>\sum_{i=1 / k e}^{n} p_{r_{i}}^{2} \operatorname{MMD}^{2}\left(\eta_{r_{i}}, \kappa_{r_{i}} ; k\right) \\\\
> > > &\dots \\\\
> > > &=\gamma^{2 \alpha} \operatorname{MMD}^{2}\left(\eta_{0}, \kappa_{0} ; k\right)
> > > \end{align}
> > > $$
> > >
> > > where the **second equality** relies on the Kernel property of MMD. This does not hold for Sinkhorn divergence. For a rigorous proof, we show that Sinkhorn divergence can **still guarantee the non-expansion on this counterexample** by leverage of its convexity[1]. We give  the detailed proof here:
> > >
> > > $$
> > > \begin{align}
> > > W_{c, \varepsilon}\left(\mathcal{T}^{\pi} \mu\left(s_{0}\right), \mathcal{T}^{\pi} \nu\left(s_{0}\right) \right) &=W_{c, \varepsilon}\left(E \left[\eta_{r}\right],E \left[\kappa_{r}\right]\right) \\\\
> > > &\leq \sum_r p_r ( \sum_{r^\prime} p_{r^\prime} W_{c, \varepsilon}(\eta_{r}, \kappa_{r^\prime})) \\\\
> > > &\leq \sum_r p_r ( \sum_{r^\prime} p_{r^\prime} W_{c, \varepsilon}(\eta_{0}, \kappa_{0})) \\\\
> > > &=W_{c, \varepsilon}\left(\eta_{0}, \kappa_{0} \right)
> > > \end{align}
> > > $$
> > > where the second inequality follows from the shift-invariant of Sinkhorn divergence.
> > >
> > > In summary, after the rigorous proof **from both sides**, you may be also curious about the behavior inconsistency between the non-expansion of Sinkhorn divergence and expansion counterexample for MMD with Gaussian kernel. We argue that ,to the best of our knowledge, both of them can be true as if there indeed exist examples in calculus where the behavior for the general case can be different from the limiting case. In order to completely figure it out, we need to dive extremely deep into the related literature, which is beyond the scope of our RL-relevant paper. We leave it as future works.
> > >
> > >
> > > [1] Feydy, Jean, et al. "Interpolating between optimal transport and mmd using sinkhorn divergences." The 22nd International Conference on Artificial Intelligence and Statistics. PMLR, 2019.

---

> > > > ### Comment · Reviewer_ptV9 · 2022-08-08
> > > > **Thank you for your response, some further questions**
> > > >
> > > > Thank you again for your response. Regarding the first set of displayed equations, I think there is still an issue with the reasoning here which means that part (3) of Theorem 1 is not correct.
> > > >
> > > > If it holds that W_{c,eps}(aX, aY) <= W_{c, eps}(X, Y), then can apply the same argument to obtain W_{c,eps}((1/a) aX, (1/a) aY) <= W_{c,eps}(aX, aY), so W_{c,eps}(X, Y) <= W_{c,eps}(aX, aY), and so combining these two inequalities, we would have W_{c,eps}(X, Y) = W_{c, eps}(X, Y) again, which is not true. For example, this would be inconsistent with the limiting case of Wasserstein distances that the authors mention in their comment. I think there is an issue in the change of variables performed in this block of equations.
> > > >
> > > > Although the authors point to examples in analysis where certain behaviours are not preserved under certain limits, I don't think this non-contraction property is such a case. If we have
> > > >
> > > > \bar{W}^\infty_{c,eps}(T^pi mu, T^pi nu) <= \bar{W}^\infty_{c,eps}(mu, nu)
> > > >
> > > > for all eps in (0, infty), as the authors claim, then we can take limits as epsilon -> infty, to obtain
> > > >
> > > > MMD_K(T^pi mu, T^pi nu) <= MMD_K(mu, nu),
> > > >
> > > > for the appropriate kernel K, which would contradict the MMD counterexample again.

---

> > > > > ### Author Response · Authors · 2022-08-09
> > > > > **Author Response: a new (close) non-expansion proof with the consistency to MMD**
> > > > >
> > > > > We sincerely appreciate your further input and efforts to improve our paper. Here we revised our conclusion to a **close non-expansion operator** for the unrectified kernel $k_\alpha$ with $a \leq 1$ with a rigorous proof as follows.
> > > > >
> > > > > Firstly, we denote $\Pi^2$ as the optimal joint distribution for $(U, V)$ and thus we write the explicit form of Sinkhorn divergence $W_{c, \varepsilon}(U, V)$ between two random variables $U$ and $V$:
> > > > >
> > > > > $$
> > > > > \begin{align}
> > > > > 	W_{c, \varepsilon}(U, V) &  = \text{KL}(\Pi^2 || \mathcal{K}) \\\\
> > > > >      &=\int_{-\infty}^{\infty}\int_{-\infty}^{\infty} \Pi^2(x, y) \log \frac{\Pi^2(x,y)}{\frac{1}{Z_2}e^{-\frac{c(x, y)}{\epsilon}} \mu(x)\nu(y)} dx dy,
> > > > > 	\end{align}
> > > > > $$
> > > > >
> > > > > where the normalization factor $Z_2$ for the Gibbs kernel $\mathcal{K}$ is $Z_2=\int_{-\infty}^{\infty}\int_{-\infty}^{\infty} e^{-\frac{c(x, y)}{\epsilon}} \mu(x) \nu(y) dx dy$ and $\mu(x),\nu(y)$ are the marginal density funtion of $U$ and $V$ with respect to $x$ and $y$.
> > > > >
> > > > > Next, we consider the scale sensitive property of Sinkhorn divergence. We also denote $\Pi^1$ as the optimal joint distribution for $(aU, aV)$. A key proof element is about the Gibbs kernel $\mathcal{K}$. By definition, the pdf of $\mathcal{K}(U, V) \propto e^{\frac{-c(x,y)}{\varepsilon}} \mu(x) \nu(y)$. After a scaling transformation, the pdf of $aU$ and $aV$ with respect to $x$ and $y$ would be $\frac{1}{a}\mu(\frac{x}{a})$ and $\frac{1}{a}\nu(\frac{y}{a})$. Thus $\mathcal{K}(2U, 2V) \propto e^{\frac{-c(x,y)}{\varepsilon}} \frac{1}{a}\mu(\frac{x}{a}) \frac{1}{a}\nu(\frac{y}{a})$. The new normalization factor $Z_1$ is formulated as
> > > > >
> > > > > $$
> > > > > \begin{align}
> > > > > 	Z_1 &=\int_{-\infty}^{\infty}\int_{-\infty}^{\infty} \frac{1}{a^2} e^{-\frac{c(x^\prime, y^\prime)}{\epsilon}} \mu(x^\prime / a) \nu(y^\prime / a) dx^\prime dy^\prime\\\\
> > > > >      &=\int_{-\infty}^{\infty}\int_{-\infty}^{\infty} e^{-\frac{c(ax, ay)}{\epsilon}} \mu(x) \nu(y) dx dy,
> > > > > 	\end{align}
> > > > > $$
> > > > > and the cost function of it is different from $Z_2$. For $\Pi^2(U, V)$, the scaled pdf of $\Pi^2(aU, aV)$ would be $\frac{1}{a^2}\Pi^2(\frac{x}{a}, \frac{y}{a})$.  Then we have the following results:
> > > > >
> > > > > $$
> > > > > \begin{align}
> > > > > 	W_{c, \varepsilon}(aU, aV) &
> > > > > 	= \text{KL}(\Pi^1 || \mathcal{K}) \\\\
> > > > > 	&\leq \text{KL}(\Pi^2 || \mathcal{K}) \\\\
> > > > > 	&=\int_{-\infty}^{\infty}\int_{-\infty}^{\infty} \frac{1}{a^2} \Pi^2(\frac{x^\prime}{a}, \frac{y^\prime}{a}) \log \frac{\frac{1}{a^2} \Pi^2(\frac{x^\prime}{a}, \frac{y^\prime}{a})}{\frac{1}{a^2}\frac{1}{Z_1}e^{-\frac{c(x^\prime, y^\prime)}{\epsilon}} \mu(\frac{x^\prime}{a}) \nu(\frac{y^\prime}{a})}  dx^\prime dy^\prime, \\\\
> > > > > 	&=\int_{-\infty}^{\infty}\int_{-\infty}^{\infty} \Pi^2(x, y) \log \frac{\Pi^2(x,y)}{\frac{1}{Z_1}e^{-\frac{c(ax, ay)}{\epsilon}} \mu(x) \nu(y) \frac{Z_2}{Z_2}}  dx dy, \\\\
> > > > > 	&=\int_{-\infty}^{\infty}\int_{-\infty}^{\infty} \Pi^2(x, y) ( \log  \frac{\Pi^2(x,y)}{\frac{1}{Z_1}e^{-\frac{c(ax, ay)}{\epsilon}} \mu(x) \nu(y)} + \log \frac{Z_1}{Z_2})  dx dy, \\\\
> > > > > 	&\stackrel{c=-k_\alpha, a\leq1}{\leq} \int_{-\infty}^{\infty}\int_{-\infty}^{\infty} \Pi^2(x, y) \log  \frac{\Pi^2(x,y)}{\frac{1}{Z_1}e^{-\frac{c(x, y)}{\epsilon}} \mu(x) \nu(y)}dx dy + \log \frac{Z_1}{Z_2} \cdot 1  , \\\\
> > > > > 	&=\mathcal{W}_{c, \varepsilon}(U, V) + \Delta(a),
> > > > > \end{align}
> > > > > $$
> > > > >
> > > > > where the second positive term $\Delta(a)=\log \frac{Z_1}{Z_2}$ satisfies $\Delta(a)\rightarrow 0$ as $a\rightarrow 1$ **(in practice $\gamma$ is very close to 1)**. The second inequality holds for the general $\varepsilon$ because for the unrectified kernel $k_\alpha=-\Vert x - y \Vert^\alpha$ **with $a \leq  1$**, for any $\varepsilon$ and $x, y$ we have
> > > > >
> > > > > $$(ax - ay)^\alpha \leq |a|^\alpha (x-y)^\alpha \leq (x-y)^\alpha  e^{-\frac{c(ax, ay)}{\varepsilon}} \geq e^{-\frac{c(x, y)}{\varepsilon}} \Rightarrow e^{-\frac{c(ax, ay)}{\varepsilon}} \mu(x) \nu(y) \geq e^{-\frac{c(x, y)}{\varepsilon}} \mu(x) \nu(y)$$
> > > > >
> > > > > However, under this condition, $Z_1 \geq Z_2$ and thus $\Delta(a)\geq 0$, but $\Delta(a)\rightarrow 0$ as $a\rightarrow 1$ (in practice $\gamma$ is very close to 1). We think there is indeed a gap between **this (close) non-expansion property of Sinkhorn divergence** and the empirical success of SinkhornDRL algorithm. The inequality is established based on the unrectified kernel, but **it is tricky to find the contractive property for Sinkhorn divergence with the Gaussian kernel for any $\varepsilon$ and $x, y$**. Thus, it is fair that some counterexamples may exist for the non-contractive $\mathfrak{T}^\pi$ under Sinkhorn divergence, which is also consistent with the counterexample MMD with Gaussian kernel (when $\varepsilon \rightarrow \infty$).
> > > > >
> > > > > In summary, we argue that although there is new proof, the (close) non-expansion conclusion is very close to previously and it is still beyond the scope of our original paper, and it does not hurt the conclusion. We sincerely hope you can take this new proof into account and have a big picture to view the contribution of our work both theoretically and empirically. Thanks very much.

---

### Official Review · Reviewer_osMq · 2022-07-11

**Rating:** 6
**Confidence:** 4
**Soundness:** 3 good
**Presentation:** 4 excellent
**Contribution:** 3 good

**Summary:**

The paper under review studied the problem of distributional reinforcement learning (DRL).
Instead of (expected) value function, DRL aim to learn the entire value distribution.
Hence to optimize (solve distributional Bellman equation),
one needs to pick a norm to measure distances between distributions.
In this paper, the authors proposed to use Sinkhorn divergence.
They proved that distributional Bellman operator under Sinkhorn divergence
interpolates between Wasserstein distance and MMD.
Moreover, they show that distributional Bellman operator is non-expansive
and approximately performs a regularized moment matching.
Furthermore, based on Sinkhorn iteration, the authors provided algorithms
for DRL with Sinkhorn divergence, and demonstrated its efficiency under Atari games.


**Questions:**

Q1: Does the choice of $\epsilon$ matter? The authors mentioned multiple times that Sinkhorn allows
a trade-off that simultaneously leverages the geometry of the Wasserstein distance ($\epsilon \to 0$) and the
unbiased gradient estimate property of MMD ($\epsilon \to \infty$).
However, in the sensitivity analysis (sec 5.2), the authors claimed that SinkhornDRL is robust to the choice of $\epsilon$ along with
other hyper-parameters.



Q2: Fig 3 are plots for the game Breakout. Do similar behaviors appear for other Atari games? For example, seaquest.
According to Fig 2, Sinkhorn shows near zero improvement compare to MMD and QRDQN (i.e. three of them behaved similarly).
Hence it seems that Breakout, the game itself is not sensitive to the choice of $\epsilon$,
or even the choice of norm (its hard to believe L=2 and L=50 give the same norm Fig 3(c)).
What is the cost function for Breakout?
maybe the cost is either close to a uniform or diagonal matrix, which makes hyper-parameters no long matter.



**Limitations:**

The authors mentioned that the main limitation of SinkhornDRL is that the superiority over existing state-of-the-art algorithms may
not be sufficiently significant. And also authors conjectured that to extend  SinkhornDRL for better performance, implicit generative
models, including parameterizing the cost function in Sinkhorn divergence shall be further incorporated.
Maybe hyperparameter tuning shall be take into consideration too.


**Strengths And Weaknesses:**

Strengths:
The paper is well written and easy to follow.
The idea of using Sinkhorn divergence is natural and interesting.
The theoretical results are nicely stated and proved.
The non-expansive on Bellman operator (Thm 1)
and Equivalence to Regularized MMD (Cor 1) have potential applications in the field.


Weaknesses:
The main concern is the motivation.

It is unclear why the authors proposed to use Sinkhorn divergence as a norm.
Because it interpolates between Wasserstein distance and MMD
or because it has properties that other existing approaches do not have?
Or it is just another norm, out preform other algos in some cases.

In the experiment section, the authors demonstrated that SinkhornDRL
is "extremely superior over existing algorithms" on some games.
The question is then what is the characterization of these games?
and why Sinkhorn is better in these cases?

More interestingly, could the authors say a bit more on
when Wasserstein is better, and when MMD is better
(probably depends on the sparsity of the OT plan).
This may lead to some guideline on choice of $\epsilon$ per game.

---

> ### Author Response · Authors · 2022-08-01
> **Response**
>
> Thank you for appreciating our work. Below we address all your concerns. Please kindly let us know if there are any further questions.
>
> ### Weakness 1: motivation of Sinkhorn divergence
> Our motivation lies in multiple and correlated perspectives. Firstly, as a norm in distributional RL, Wasserstein distance, albeit being tricky to directly evaluate, exhibits huge empirical success e.g., QR-DQN. Sinkhorn divergence, which is renowned as a successful norm to approximate Wasserstein distance, has not been studied in distributional RL setting. Secondly, Sinkhorn divergence enjoys intriguing theoretical properties interpolating between Wasserstein and MMD, and the latter has recently achieved state-of-the-art performance. Thus it is natural to explore the efficacy of  ``intermediate'' Sinkhorn divergence in the distributional RL setting. Finally, sinkhorn divergence has been successfully applied in the generative models, and researchers are definitely curious whether we can apply this divergence to another important research field where the distributional divergence serves as one of the key factors for algorithm design, e.g., distributional RL.
>
> ### Weakness 2: explanation of Sinkhorn's outperformance in certain games
> This is a very interesting yet open question in the RL community as to the best of our knowledge, existing (distributional) RL works have not studied what kind of environments are more suitable for a specific class of distributional RL algorithms, e.g., QR-DQN and MMDDRL. Thus, the common choice in research papers is to provide a ratio improvement barplot across all Atari games and compare the mean or median performance of them. In terms of the superiority of Sinkhorn divergence in certain games, one possible explanation is that SinkhornDRL can both enjoy the geometry property of Wasserstein distance and unbiased gradient estimate for MMD, and this nice trade-off can potentially yield remarkable superiority in certain games, where the value distribution might be more complicated to represent and reducing gradient bias can be substantially helpful to the optimization.  We leave more rigorous investigation towards this issue in the future.
>
> ### Weakness 3: MMD or QR-DQN for each game
> Related to Weakness 2, there is less study about in which environment MMD or Wasserstein distance can be better or not. We think it really depends on the game or environment. For convenience, we uniformly choose $\varepsilon=10.0$ after doing the sensitivity analysis in breakout, but it is definitely well-expected that a careful sweep over $\varepsilon$ for each game can achieve better performance.
>
> ### Question 1: choice of $\varepsilon$
> As shown in the sensitivity analysis part, the performance of SinkhornDRL is robust to $\varepsilon$ in a certain range, e.g., [1, 500] for breakout, and thus we uniformly choose $\varepsilon=10.0$ across all Atari games.
>
> Also, as shown in Theorem 1, in the ideal case when we ignore the approximation errors and optimization difficulties, the SinkhornDRL’s performance should tend to Wasserstein distance-based algorithm, e.g., QR-DQN, if we decrease $\varepsilon$ and should tend to MMDDRL if we increase $\varepsilon$. To demonstrate that, we also supplemented results in Figure 10 Appendix F of the revised paper and show that SinkhornDRL approaches QR-DQN with decreasing $\varepsilon$ and gets close to MMDDRL with increasing $\varepsilon$. These empirical results coincide with our theoretical analysis.
>
> Meanwhile, it is also important to note that an overly large or small $\varepsilon$ will lead to the numerical instability issue. An overly small $\varepsilon$ will lead to a trivial almost 0 $\mathcal{K}$ in Sinkhorn iteration in Algorithm 2, and will cause $\frac{1}{0}$ numerical instability issue for $a_l$ and $b_l$ in Line 5 of  Algorithm 2. An overly large $\varepsilon$ will let the $\mathcal{K}_{i, j}$ explode to $\infty$.
>
> In practice, a proper $\varepsilon$ is suggested because it can avoid the numerical instability issue analyzed above and achieve robust performance demonstrated in our sensitivity analysis part simultaneously.
>
> ### Question 2: sensitivity analysis on Seaquest
> We supplemented the results on Seaquest in Figure 11 of Appendix F of the revised paper. It turns out that the robust performance is also applied in Seaquest when $\varepsilon=10,100, 500$. Meanwhile, SinkhornDRL with a small $\varepsilon=1$ is slightly worse than others and tends to QRDQN, which is also consistent with our theoretical result in Theorem 1.

---

> > ### Comment · Reviewer_osMq · 2022-08-09
> > **Thank you for your response.**
> >
> > Thank you for your response, here are some follow up questions.
> >
> > Q1 For figure 3(c), breakout presented similar behavior for L = 2 and L = 50. It means Sinkhorn basically converged in 2 steps?
> > What is the cost matrix corresponding to breakout? is it almost uniform?
> >
> > Q2 For Seaquest, besides varying $\epsilon$, is it robust to the choice of L and sample size too as breakout  (Fig 3 (b)-(c))?
> >
> > Q3 For Seaquest, when $\epsilon =1$, it starts to tend to QRDQN. How big a $\epsilon$ needs to be , for it to start behave like MMDDRL?
> > One would think $\epsilon = 500$ is already sufficiently large, but it does not see like the case here.

---

### Official Review · Reviewer_4n2i · 2022-07-11

**Rating:** 6
**Confidence:** 2
**Soundness:** 3 good
**Presentation:** 3 good
**Contribution:** 3 good

**Summary:**

This paper proposed a distributional RL algorithm that based on Sinkhorn divergence. It derived the theoretical guarantee for Sinkhorn divergence with the distributional Bellman operators and evaluated empirical performances on Atari games.


**Questions:**

1. It would be great to have more justifications on why the 2nd term on eq(12) will goes to 0 when $\epsilon\rightarrow\infty$ since the multiplication of $\infty$ and $\log(1)$ can be undetermined.
2. Why not using the same Gaussian kernel for MMD and SinkhornDRL to validate their theoretical relations by changing $\epsilon$?
3. Is the conclusion from Figure 2 (a) and Figure 3 (a) not super aligned with each other? Figure 2(a) demonstrates that a Sinkhorn divergence in DRL is better than QRDQN with Wasserstein distance, while Figure 3 (a) shows that a smaller $\epsilon$ doesn't affect the performance much. I didn't fully get it because the Sinkhorn divergence with a small $\epsilon$ will degenerate to Wasserstein distance as shown in Theorem 1.


**Limitations:**

The authors have addressed the major limitation of this paper with potential future directions.

**Strengths And Weaknesses:**

Strengths:

This paper is well-written and easy to follow, and it is interesting to learn the connections between different approaches from a unified framework.

Weakness:
I have two major concerns about this paper:
1. The novelty of this paper. The relationship among Wasserstein distance, MMD and Sinkhorn divergence has been studied in existing literatures [1].
The discussion of the Sinkhorn divergence and MMD and a Gaussian kernel looks very interesting, however, it seems unrelated to the experiment part given that SinkhornDRL is not using a Gaussian kernel.

2. The empirical performances (Figure 1 and Figure 4) of SinkhornDRL is not strong enough to demonstrate SinkhornDRL outperforms the other algorithms, where MMD and SinkhornDRL perform similarly on 10/18 tasks.

[1]: Genevay, Aude, Gabriel Peyré, and Marco Cuturi. "Learning generative models with sinkhorn divergences." International Conference on Artificial Intelligence and Statistics. PMLR, 2018.

---

> ### Author Response · Authors · 2022-08-01
> **Response**
>
> Thanks for your valuable suggestions. Below we address all your concerns. Please kindly let us know if there are any further questions.
>
> ### Weakness 1: novelty issue
> Our SinkhorndRL algorithm not merely depends on relationships of divergences, but provides a deeper investigation and successful application in the RL scenario as opposed to generative models in [1]. Firstly, distributional RL is a vastly different setting compared with generative models [1], and our successful application in the RL scenario can potentially inspire more research directions in the future. Moreover, the design of GAN in [1] may directly rely on relationships of divergence to measure the distance between two distributions, while our SinkhorndRL further probes the convergence (rates) of distributional Bellman operators under these divergences as suggested in Theorem 1. More importantly, the relationship between MMD and Sinkhorn divergence for the general $\epsilon \in (0, \infty)$ is further explored in Proposition and Corollary 1, which goes much farther compared with [1]. Also, we clarify that the relationship between MMD and Sinkhorn in Corollary 1 does not require the Gaussian kernel.
>
> ### Weakness 2: empirical significance
> We randomly pick games in Figures 1 and 4 from 55 Atari environments without cherry-picking. The empirical fact is that there is no algorithm that can outperform the others across all Atari games, and thus it is suggested to compare performance from a more comprehensive viewpoint in Figure 2. Particularly, SinkhornDRL is very competitive with the SOTA algorithm in each game and remarkably outperforms others in a large number of games.
>
> ### Q1: justification
> Here we give a detailed justification. The key point is to show that $\varepsilon_{k}  \mathrm{KL}\left(\Pi_{k}, \mu \otimes\nu \right) \rightarrow \infty \times 0 = 0$. To show that, based on Line 446, we have the following inequality
>
> $\int c(x, y) \mathrm{d} \Pi_{k}+\varepsilon_{k} \mathrm{KL}\left(\Pi_{k}, \mu \otimes \nu \right) \leq \int c(x, y) \mathrm{d} \mu \otimes \nu+0$ (when $\Pi(\mu, \nu)=\mu \otimes \nu$ as a special case).
>
> We have shown that $\Pi_{k} \rightarrow \mu \otimes \nu$ as $\varepsilon_k \rightarrow \infty$, and thus in this limiting scenario, the inequality above becomes
>
> $\int c(x, y) \mathrm{d} \mu \otimes \nu+\varepsilon_{k} \mathrm{KL}\left(\Pi_{k}, \mu \otimes \nu \right) \leq \int c(x, y) \mathrm{d} \mu \otimes \nu$
>
> Since $\mathrm{KL}\left(\Pi_{k}, \mu \otimes \nu \right)$ is non-negative, and the unique minimizer $\Pi_{k}$ exists, it must be the case that $\varepsilon_k \mathrm{KL}\left(\Pi_{k}, \mu \otimes \nu \right) \rightarrow 0$ to hold the inequality above.
>
> ### Q2: Relationships by varying $\varepsilon$
> We can vary $\varepsilon$ in a proper range, but an overly large or small $\varepsilon$ will lead to the numerical instability issue. An overly small $\varepsilon$ will lead to a trivial almost 0 $\mathcal{K}$  in Sinkhorn iteration Algorithm 2 and will cause $\frac{1}{0}$ numerical instability issue for $a_l$ and $b_l$ in Line 5 of  Algorithm 2. An overly large $\varepsilon$ will let the $\mathcal{K}_{i, j}$ explode to $\infty$. This also leads to numerical instability. Therefore, the ideal scenario when $\varepsilon$ is sufficiently small or large may not be fully implemented, and thus there indeed exists a slight gap between theory and experiments.
>
> However, we can still characterize the trend of SinkhornDRL to approach MMDDRL by carefully increasing $\varepsilon$, and we supplemented this result in Figure 10 (as well as the large $\varepsilon$ case to compare with QR-DQN) of Appendix F in the revised version. It turns out that SinkhornDRL tends to MMDDRL as we increase $\varepsilon$ from 500 to 1,000 and 5,000.
>
> The reason why we do not use Gaussian kernel is that we find SinkhorndRL with an unrectified kernel $\alpha=$ 1 or 2 works better than Gaussian kernel in distributional RL setting, which additionally requires carefully tuning the bandwidth. Note that the theoretical relationship between MMD to SinkhornDRL (except the moment matching part) does not require the Gaussian kernel and MMD with an unrectified kernel can still work well as shown in the original MMDDRL paper.
>
> ### Q3: Figure 2(a) and Figure 3(a)
> They are aligned with each other. Figure 3(a) shows that the performance of SinkhornDRL is robust to $\varepsilon$ within a certain range, e.g., [1, 500]. Meanwhile, if we further decrease $\varepsilon$ to 0, the performance of SinkhornDRL will tend to QR-DQN. We supplemented this result in Figure 10 of Appendix in the revised version to compare QR-DQN. It should be also noted that the extreme case can not be implemented due to the fact that an overly large or small $\varepsilon$ will lead to the numerical instability issue, but we empirically show that the trend still holds, which coincides with our theoretical results. The analysis of MMD between Figure 2(b) and Figure 3(b) is similar.

---

### Author Response · Authors · 2022-08-02
**Response to all reviewers**

Thank you very much for all reviewers’ valuable inputs to our paper.

We have responded to each reviewer about all of their concerns, respectively. We hope our response and clarification could help you better understand our technical details and contribution. **Also, we summarized the main changes in the updated version of our paper**.

1. We added more **sensitivity analysis experiments by varying $\varepsilon$ on both breakout and Seaquest games in Figures 10 and 11 of Appendix F**. It shows the performance of SinkhornDRL is still robust with $\varepsilon$ in a certain range, e.g., [1, 500] and SinkhornDRL tends to QR-DQN and MMD if we decrease and increase $\varepsilon$, respectively. This is consistent with our theoretical result in Theorem 1. In addition, we also point out that an overly small or large $\varepsilon$ is likely to cause the numerical instability issue in Sinkhorn iterations. These additional sensitivity experiments are more helpful to guide us to choose $\varepsilon$ in the practical deployment of our SinkhornDRL algorithm.

2. We also added an ablation study on the **computational cost in Figure 13 of Appendix F** regarding the number of samples, and thus our sensitivity analysis is more comprehensive and convincing.

3. We revised **typos and some inaccurate claims** as suggested by some reviewers and the revised paper could be more accurate and convincing.

Please kindly let us know if you have any further questions and we are very happy to respond to any further concerns.

Yours faithfully,

Authors

---

### Meta-Review · Area_Chair_iV7R · 2022-08-27

**Recommendation:** Reject
**Confidence:** Certain

**Metareview:**

All reviewers acknowledged that this paper is an interesting contribution to the distributional RL literature. Some concerns brought up by reviewers in their initial reports included issues with the statement of Theorem 1, a non-standard evaluation protocol for the Atari experiments, and that some claims were overly strong, such as the robustness claim of Figure 3.

During the discussion, reviewers found that the rebuttal addressed a few of these concerns, but that a few concerns still remained.
While the claims of Theorem 1 have been revised significantly, and the reviewers appreciated this. However, the clarify of the statement is still not clearly presented, and the reviewers could not verify the correctness of the proof in its current form. For instance, one reviewer pointed out that"T^pi is a closely non-expansive operator" is not a precise mathematical statement and is not clearly defined in the paper. Another point was that the proof could be more clear and therefore it was difficult to verify its correctness, in addition to a number of typos: (Eqns (39), (40), 1/Z_1 should be 1/Z_2; equation below Line 518 is missing an implication sign; Line 510 K(2U,2V) should be K(aU, aV)).

Several other reviewers were not convinced by the empirical results in that they felt like the paper would be stronger if (1) it used the same evaluation protocol as other distributional RL papers and (2) results on the behavior of the algorithm when sweeping over key hyperparameters (epsilon, cost choice, L), would also considerably strengthen the paper by clarifying how the method works in practice.

Overall, reviewers found the contribution to be promising and a step in the right direction. After another revision addressing these issues, I think the paper will be a strong contribution to the distributional RL literature.


**Award:**

No

---

### Decision · Program_Chairs · 2022-09-14

Reject